



# A method to use proxy data of runoff-related impacts for the evaluation of a model mapping intense storm runoff hazard: application to the railway context

**Isabelle Braud[1], Lilly-Rose Lagadec[2,1,3], Loïc Moulin[3], Blandine Chazelle[4], and Pascal Breil[1]**

[1]INRAE, RiverLy, 5 Rue de la Doua, CS 20244, 69625, Villeurbanne, France
[2]SNCF Réseau, Engineering and Projects South-West, PIEG General Studies, 54 bis rue Amédée Saint Germain, 33077 Bordeaux, France
[3]SNCF Réseau, Engineering and Projects Direction, Railways, Tracks & Environment Department 6 avenue Francois Mitterrand, 93210 La-Plaine-Saint-Denis, France
[4]SNCF Réseau, Engineering and Projects South-East, PIEG General Studies, 31 Avenue Albert-et-Elisabeth, 63037 Clermont-Ferrand CEDEX, France

**Correspondence:** Isabelle Braud (isabelle.braud@inrae.fr)

**Abstract.** The IRIP method, or "indicator of intense pluvial runoff" in English, is a geomatics method that allows mapping the susceptibility of a territory to surface runoff and that provides three maps of susceptibility to the generation, transfer and accumulation of runoff. It is based on the combination of binary maps that represent the impact of a given factor (favourable or not favourable) on runoff. These factors are summed up to provide susceptibility maps for runoff with levels ranging from 0 to 5. To be used for risk prevention, the quality and limitations of the produced maps must be assessed. However, direct runoff data are very scarce and not available everywhere in a territory. Proxy data of impacts related to runoff can provide information useful for the evaluation of the IRIP maps. However, both pieces of information cannot be compared directly, and a specific methodology to compare susceptibility maps and proxy data must be proposed. This paper presents such a method, which accounts for the hazard level, the vulnerability of the study area and possible mitigation actions taken to reduce the risk. The evaluation method is assessed using a comprehensive database of runoff-related impacts collected on an 80 km railway line in Normandy (north of France) and covering the whole 20th century. The results show that the evaluation method is robust, relevant and generic enough for evaluating a non-quantitative method of runoff hazard mapping us-

ing localized runoff-related proxy data. In addition, the good performance of the IRIP model in the case study confirms that the susceptibility maps produced by the IRIP model provide relevant information related to runoff and that they can be used to design risk management strategies, as illustrated in the railway context.

## 1 Introduction

Runoff occurring outside of the river network is a natural hazard that is often quite localized but has a high societal impact. In France, Moncoulon et al. (2014) mention that about half of insurance claims due to flooding occur outside areas mapped as at risk of river flooding in the framework of the European Union Floods Directive. Runoff consequences can be fatalities, damage to buildings or infrastructures, or disruption of transport networks. In addition, surface runoff events are often associated with sediment transport and deposition, causing losses to agricultural land and increasing the damage to infrastructures. Linear transport networks, such as railways, are very sensitive to runoff hazards: they cross various small catchments, and water and mud can damage the railway track and electric installations (Chazelle et al., 2014; Lagadec et al., 2018). Maurer et al. (2012) estimated that the

median cost (direct and indirect) of a hydrometeorological event on the European railway network was EUR 2.69 million.

As runoff can occur everywhere on a territory, there is a need to provide maps of susceptibility to surface runoff at the scale of a whole territory or an entire transport network. Physically based distributed models may be deployed (e.g. Dabney et al., 2011; Le Bissonnais et al., 2002; Schmocker-Fackel et al., 2007; Smith et al., 1995). They have the ability to provide the spatial and temporal evolution of runoff dynamics (water depth and sometimes velocity). However, they require many input data for their set-up and calibration that may not be available everywhere. Thus, this kind of model may be difficult to deploy in large territories. An alternative solution, called IRIP, or "indicator of intense pluvial runoff", was proposed by Dehotin and Breil (2011) for mapping the susceptibility to surface runoff. The IRIP model allows the creation of three maps documenting three different phases of the surface runoff phenomenon: generation, transfer and accumulation. It is based on a score method using a set of indicators derived from easily available information (digital terrain model, land use map and soil map). The result is composed of three susceptibility maps with scores ranging from 0 (no susceptibility) to 5 (high susceptibility) for the generation, transfer and accumulation of runoff. IRIP maps are static; therefore, the IRIP model does not have any temporal resolution, and the maps do not provide quantitative information on runoff dynamics. However, the maps remain useful for prevention purposes, provided they are properly evaluated. Thus, the IRIP model and its evaluation are the focus of this paper.

Indeed, to be used for hazard prevention and risk management, the validity of the produced maps must be assessed and the limits of the methodology clearly defined. However, runoff located outside of the river network is a phenomenon that is difficult to observe, as it can occur everywhere and over very short durations. There are therefore very few direct observations of runoff, apart from artificial runoff simulation experiments or some rare research experiments (see Dehotin et al., 2015, for more details). On the other hand, indirect information on runoff-related impacts can be more easily available, as runoff may have damaging consequences such as flooding of buildings or of transport networks (roads or railways), mudflows, erosion, or landslides. Information on these impacts can be collected and reported based on various media: post-event surveys to collect the location of impacts on infrastructures or on transport networks (Versini et al., 2010b; Naulin et al., 2013; Defrance et al., 2014; Lagadec et al., 2016b, 2018), insurance claims on buildings or infrastructures (Moncoulon et al., 2014; Le Bihan et al., 2017), analyses of the press and social media (Llasat et al., 2013; Saint Martin et al., 2018; Petrucci et al., 2019), or citizen science (Gourley et al., 2010; Le Coz et al., 2016). All these data are referred to as "proxy data" in this paper. Such data have been used for the evaluation of quantitative flash-flood forecasting models (e.g. Gourley et al., 2010; Defrance et al., 2014; Javelle et al., 2014; Saint-Martin et al., 2016), road cutting warning models (Versini et al., 2010b; Naulin et al., 2013) or flooding impact models on buildings (Le Bihan et al., 2017). The evaluation is based on criteria that are used for the evaluation of meteorological or hydrological forecasts (WWRP/WGNE Joint Working Group on Forecast Verification Research, 2015), i.e. the computation of probability of detection (POD), the false-alarm ratio (FAR) and the success ratio. Such an approach has been extended for the evaluation of non-quantitative prediction models such as the IRIP model by Lagadec et al. (2016b). It was further improved by Lagadec et al. (2018) using a comparison with expert judgement, taking into account the vulnerability of the railway. However, these evaluations remained qualitative. It was necessary to generalize the evaluation methodology and to propose a more systematic and quantitative manner to deal with proxy data of localized runoff-related impacts. It was also necessary to use a large and comprehensive data set of runoff-related impacts to assess the relevance and robustness of the proposed evaluation methodology and to highlight the limitations of the IRIP maps before their use in risk management strategies.

Indeed, although proxy data provide useful information on the occurrence of runoff, these data cannot be compared directly to susceptibility maps of runoff hazards because the information they carry is not the same. Impacts are related to the occurrence of a risk, thus taking into account the vulnerability of the stakes (for instance an infrastructure will be less vulnerable to runoff if there is a protection structure), whereas susceptibility maps only describe hazards. Susceptibility maps to runoff are continuous in space, whereas proxy data are generally point data (impacts on buildings or transport networks). Impacts are generally observed where there are stakes. So the information may not be comprehensive, in particular when runoff occurred without stakes. This comprehensive information would be required to accurately estimate the false-alarm ratio (Calianno et al., 2013). In addition, proxy data are not always well geolocalized, and the description of impacts is subjective and depends on the observer.

The paper focuses on one kind of proxy data that are localized runoff-related impacts, such as impact on transport networks. The objective of the paper is to propose a methodology to use these proxy data for the evaluation of a non-quantitative method of runoff hazard mapping, such as the IRIP method. This implies identifying which data and information processing methods are required to perform such an evaluation and the criteria that can be used for a quantitative comparison. Then, the feasibility and relevance of the proposed methodology are assessed using a well-documented case study in the railway context. The study takes advantage of the availability of databases of damage and incidents on the French railway network. The case study is a particularly well-documented 80 km railway between Rouen and Le Havre in northern France, where the IRIP model was set

up and where a comprehensive database of runoff-related impacts has been collected for about 1 century. This provided a comprehensive proxy data set that allowed the assessment of the robustness and applicability of the proposed evaluation methodology. The data set also allowed the assessment of the relevance of the IRIP model for runoff hazard mapping on a wide area in the railway context.

The paper is organized as follows. The "Materials and methods" section presents the IRIP model, the proposed evaluation methodology, the case study, and how the IRIP model and the evaluation methodologies were set up in the case study. Then the "Results" section presents the results of the IRIP model and those of the evaluation on the 80 km railway between Rouen and Le Havre. In the "Discussion" section, we discuss the relevance of the evaluation method, its sensitivity to the data accuracy and model set-up, and the genericity of the proposed methodology. The use of the IRIP model for risk assessment in the railway context is also discussed before providing the main conclusions of this study.

## 2 Materials and methods

### 2.1 The IRIP mapping model

The IRIP model is briefly described here, but more details can be found in the literature (Dehotin and Breil, 2011; Lagadec et al., 2018). The present description is mainly taken from Lagadec et al. (2018), which retained improvements proposed by Lagadec (2017) to the IRIP model. The IRIP model provides three maps representing three processes involved in storm runoff hazard: generation, transfer and accumulation of runoff. Runoff generation occurs in areas with low infiltration capacity, shallow soils or saturated soils, leading to runoff produced by infiltration excess and/or saturation excess. Runoff transfer occurs in areas where water can be transferred downwards, can be accelerated and can induce erosion, depending on soil erodibility. Runoff accumulation occurs in areas where water can slow down, concentrate and be accumulated to produce floods and sediment load deposits. The IRIP model focuses on runoff occurring outside of the river network. It is therefore complementary to flooding risk mapping along river networks. Each IRIP map is produced by combining five indicators derived from geographic information layers (Fig. 1; Table 1). Each indicator is classified into two categories: not favourable to runoff, where 0 is attributed to the pixel, or favourable to runoff, where 1 is attributed to the pixel. This yields five binary maps that are then added to create a susceptibility map with six levels, from 0 (not susceptible) to 5 (very susceptible). The indicators used for producing each of the three susceptibility maps are presented in Fig. 1. The generation map is produced using one indicator derived from a land use map, one indicator derived from the topography and three indicators derived from a soil map. The indicator related to topography is a combination

of the slope and the topographic index (Beven and Kirkby, 1979) and is assigned the level of 1 if both are favourable and 0 if one is not favourable. The generation map is then considered to be one of the input indicators for the two other maps of susceptibility to transfer and accumulation. This allows accounting for the need of significant runoff generation to increase the susceptibility to runoff transfer and/or accumulation. Maps of susceptibility to transfer and accumulation of runoff are produced using mainly indicators based on topography. But the indicators have opposing conditions for being favourable to runoff. For instance, the slope indicator is favourable for transfer in the case of steep slopes and for accumulation in the case of low slopes. The break of the slope indicator is favourable for transfer in the case of a convex break of slopes and for accumulation in the case of a concave break of slopes. Topographic indicators are computed for each pixel relative to their upstream sub-catchment, allowing accounting for upstream-to-downstream water transfer. The resolution of the susceptibility maps retains the resolution of the digital elevation model (rasterized topography map) used as input data. To determine the thresholds separating the topographic indicator values (slope and topographic index, respectively) into values favourable or not favourable to runoff, an automatic classification, the $k$-means clustering method for grids (http://www.saga-gis.org/saga_tool_doc/2.2.5/imagery_classification_1.html [TS1]) provided in SAGA GIS (http://www.saga-gis.org/en/index.html [TS2]; System for Automated Geoscientific Analyses geographic information system), was used. The third option of the function that combines two methods, the iterative minimum distance (Forgy, 1965) and the hill-climbing method (Rubin, 1967) to divide the grid values into two classes, was used. The principle of the method is to maximize the inter-class variance while minimizing the intra-class variance. As the classification is performed using all the grid points located in the study area, the threshold value, separating the two classes (favourable or not favourable to runoff), depends on the study area. The IRIP model can therefore be applied to various territories without a priori local knowledge on the area, as the thresholds can be automatically computed. If local knowledge on threshold values is available, the user can alternatively specify these threshold values.

### 2.2 The evaluation framework

The proposed evaluation framework is shown in Fig. 2. It extends the work of Lagadec et al. (2016b, 2018) but remains based on the use of contingency tables and the computation of a detection rate and a false-alarm ratio (see details below) to propose a quantitative comparison between the IRIP maps and the localized runoff-related impacts. The method takes into account the following elements: the different nature of the impacts (localized) and the IRIP maps (continuous score maps), the vulnerability of the stakes for which runoff-related impacts are reported, and the existence of mitigation mea-

**Table 1.** Parameterization of the IRIP model for the case study. The table provides values of the thresholds used for each indicator when condition is favourable (score of 1).

| IRIP maps | Indicators | Thresholds used for favourable conditions (score $= 1$) |
|---|---|---|
| Generation | Soil permeability | Saturated hydraulic conductivity $(K_s) < 10^{-6}\,\mathrm{m\,s^{-1}}$ + urban areas |
| | Soil thickness | Soil thickness $< 50\,\mathrm{cm}$ + urban areas |
| | Soil slacking | Urban areas + slacking $\geq 3$<br>Slacking computed according to Cerdan et al. (2002) |
| | Topography | Slope $>$ Threshold_1 *or* topographic index $>$ Threshold_2<br>Threshold_1 and Threshold_2 determined using a classification algorithm (Rubin, 1967) |
| | Land use | Urban areas and agricultural lands |
| Transfer | Upstream generation susceptibility | Modal value of the upstream sub-catchment $\geq 3$ |
| | Slope | Slope $>$ Threshold_1 |
| | Break of slope | Convex break of slope $\geq 0.0018$<br>(GRASS GIS r.param.scale function; three pixels) |
| | Drained area | Drained area $\geq 2.5\,\mathrm{ha}$ (Lagadec, 2017) |
| | Soil erodibility | Erodibility – urban areas $\geq 3$<br>Erodibility computed according to Cerdan et al. (2002) |
| Accumulation | Upstream generation susceptibility | Modal value of the upstream sub-catchment $\geq 3$ |
| | Slope | Slope $\leq$ Threshold_1 |
| | Break of slope | Concave break of slope $\leq -0.0018$<br>(GRASS GIS r.param.scale function; 3 pixels) |
| | Topographic index | Topographic index $>$ Threshold_2 |
| | Drained area | Drained area $>$ 2.5 ha (Lagadec, 2017) |

sures that may reduce the occurrence of risk. The four steps of the method are detailed below.

### 2.2.1   Step 1: definition of the evaluation area

The IRIP maps can be computed over a whole territory. The evaluation area, i.e. the area where quantitative measures are computed, must be relevant to the available runoff-related impact data. This is particularly important to get a reliable estimate of false-alarm ratio. The evaluation area will therefore depend on the runoff-related impact database, as illustrated by the following examples. In case of impacts following a specific localized rainfall event, the evaluation area may be defined as the area experiencing rainfall larger than a specified intensity (see discussion about the choice of the threshold in Sect. 4.3), as if there is no rain, there is no runoff. In the case of a transport network, the IRIP maps are established for all the catchments that are intercepted by the transport network. If impacts are only recorded on the transport network, the evaluation area will be the transport track itself, with a buffer zone consistent with the resolution of the DTM used to compute the IRIP maps. This buffer accounts for in-

accuracy in the DTM and in the geolocalization of the impact data. When a comprehensive database of runoff-related impacts is available over a territory, for a long historical period, it can be assumed that the entire territory may have been affected by a rainfall event, and the entire catchment can be considered to be the evaluation area for the application of the evaluation method.

### 2.2.2   Step 2: characterization of the vulnerability and hazard in the evaluation zone

The IRIP model provides susceptibility maps with score values ranging from 0 (no susceptibility) to 5 (high susceptibility). To compare these scores with runoff-related impact data it is necessary to choose which levels of susceptibility computed by the IRIP model will generate a situation at risk. In this study, the risk is defined by combining a susceptibility level to a vulnerability level based on the exposure and known consequences of overland runoff on the railway elements. As the IRIP model provides three maps, it also means choosing the maps that will be considered in the evaluation. Previous experience (Lagadec et al., 2016b, 2018) showed

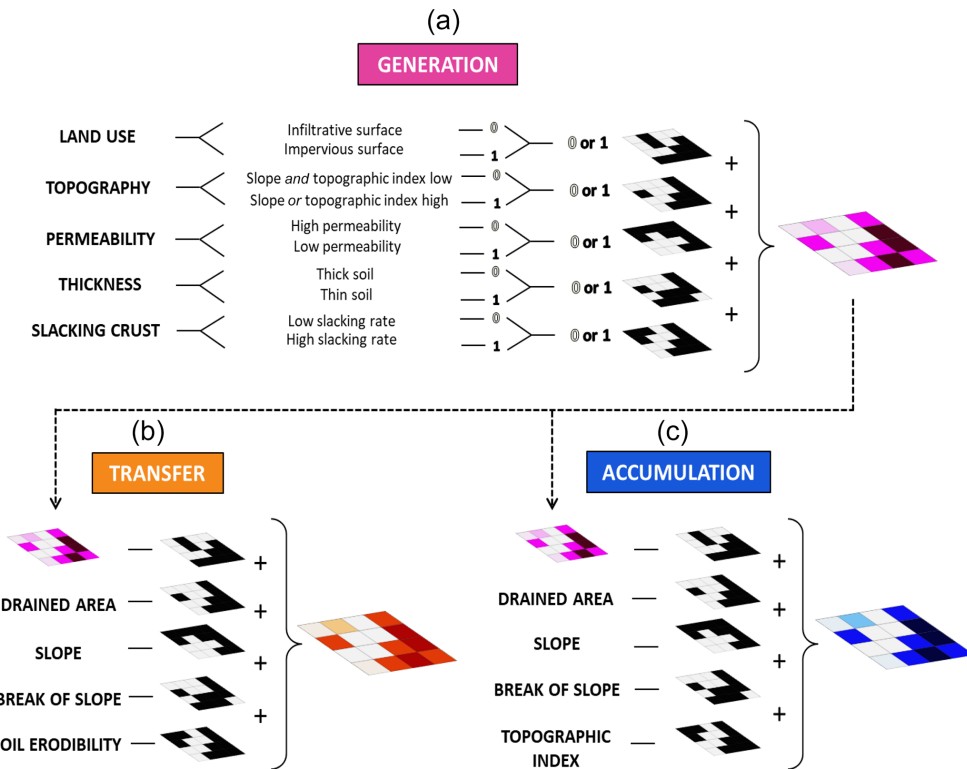

**Figure 1.** Scheme of the IRIP model presenting the various indicators computed to produce the susceptibility maps to runoff generation **(a)**, transfer **(b)** and accumulation **(c)** (adapted from Lagadec et al., 2018).

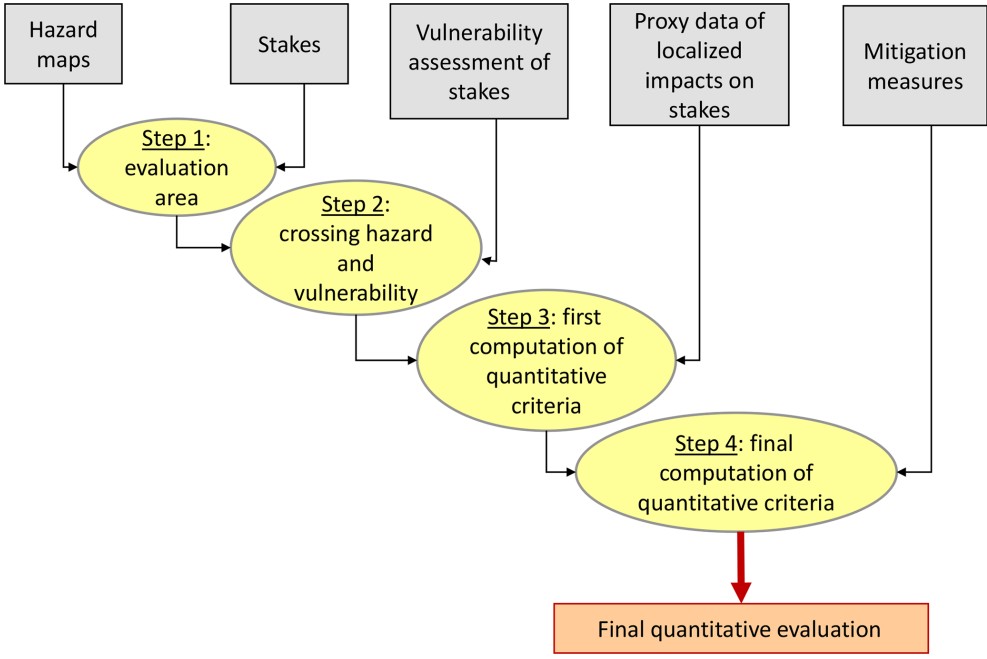

**Figure 2.** Scheme of the evaluation methodology to assess the relevance of susceptibility maps to runoff, using localized runoff-related impact proxy data. The grey boxes indicate the information that is used in the various steps of the methodology. Yellow circles present the various steps of the evaluation methodology leading to the final quantitative evaluation (orange box).

that, when compared to localized impact data, susceptibility maps for transfer and accumulation were relevant, with the susceptibility map for transfer generally associated with erosion, and the susceptibility map for accumulation associated with sediment deposition and flooding. On the other hand, localized runoff-related impact data are not directly related to the runoff generation process. In such conditions, the susceptibility map to runoff generation cannot be used for characterizing the hazard level, and a composite of the susceptibility maps to runoff transfer and accumulation is used to define the hazard.

In addition, the notion of risk depends on the exposure and vulnerability of the stakes that are considered. Several methods can be used to assess the vulnerability of stakes. Saint-Martin et al. (2016) used the AHP (analytic hierarchy process; Saaty, 1990) method to rank several stakes. Another possibility is to use a vulnerability tree based on expert judgement. For buildings, a vulnerability value can be allocated to each building and the hazard level computed in a buffer zone around the building. In the case of a transport network, it is necessary to divide the network into sections that are meaningful to runoff risk and to assign a vulnerability value to each section. The hazard level must also be computed in each section. As a buffer zone around an impact or a transport network section contains several IRIP pixels with different values of susceptibility, a rule must be chosen to assign a susceptibility value to the buffer area or the section. For instance, the hazard level can be computed as the maximum value of the susceptibility inside the buffer area or the transport network section. This accounts for the uncertainty in damage location and the observed possibility that intense runoff can flow along transport network before a damage.

Once established, the vulnerability scale is converted to a limited number of vulnerability classes. These are then combined with the hazard levels to define which combinations are at risk. The notion of the area at risk is defined according to the principle in which "the higher the vulnerability, the lower the hazard level triggering a risk". Figure 3 provides examples of such choices. Figure 3a corresponds to a case where the vulnerability of the stakes is not taken into account and where IRIP hazard levels 4 and 5 indicate being at risk everywhere. Figure 3b and c correspond to two different ways of combining hazard and vulnerability.

### 2.2.3   Step 3: quantitative evaluation of the maps

The third step is the comparison between the areas identified as being at risk in the previous step and runoff-related impact data. For this purpose, a contingency table (Table 2) is built for a sample containing all the buffer areas or sections for which a risk level has been assigned in step 2. If an impact has been observed in an area considered at risk, the impact is counted as a "hit". If no impact has been observed in an area not considered at risk, the impact is counted as a "correct negative". If an impact has been observed and the area

**Table 2.** Contingency table.

|                          | Observed impact | No observed impact |
| ------------------------ | --------------- | ------------------ |
| Area declared at risk     | Hits            | False alarms       |
| Area declared not at risk | Misses          | Correct negatives  |

is not declared at risk, the impact is counted as a "miss". Finally, if no impact has been observed but the area is declared at risk, the impact is counted as a "false alarm". Based on the contingency table, three quantitative measures of performance are computed (Table 3): the POD, which represents the fraction of impacts that have been correctly identified in an area at risk. The FAR indicates the proportion of areas at risk with false alarms. If the method were perfect, the POD would be equal to 1 and the FAR to 0. The $\chi^2$ test is used to define if the dependency between risk levels and the occurrence of impacts is significant. For that, the $\chi^2$ is compared to that of the theoretical distribution with full independence of risk and impacts. For a contingency table with 1 degree of freedom (as in our case), the probability to get a $\chi^2$ larger than 10.83 is lower than 0.1 %. Thus, a value of $\chi^2$ larger than 10.83 means that the null hypothesis (independence between the risk levels and the IRIP map) can be rejected at the 0.1 % level.

### 2.2.4   Step 4: taking into account risk mitigation measures

This fourth step is necessary to properly take into account the fact that, if an area is at risk, the stakeholder may have taken mitigation measures that may explain the absence of observed impact. Such mitigation measures are therefore likely to explain a certain number of false alarms. For instance, mitigation structures can be protection to buildings, retention basins, hydraulic works crossing below transport infrastructures, etc. They can also be resilience actions like reinforced supervision in case of high-rainfall-amount warning. Their aim is to reduce damage consequences by issuing early warnings or by performing local work to reduce potential damage during an event. If an area classified as at risk has a specific supervision measure or mitigation structures have been built, it is moved from false alarm to hit, as the implementation of mitigation measures means that the area was indeed at risk but that no impact was recorded due to the efficiency of the mitigation measure. This step will be referred to as step 4.1 in the following. If mitigation measures can be considered to be a reliable source of information regarding runoff risk, the section must also be moved from a correct negative to a miss if a mitigation measure is present and the area was not classified as at risk (this step will be referred as step 4.2 in the following). The performance measures are then recomputed

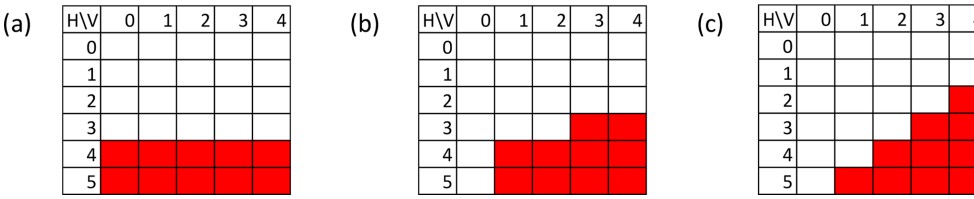

**Figure 3.** Examples of combination of hazard (H) (vertical) and vulnerability (V) (horizontal) to define the areas at risk with regards to runoff (red cells) when **(a)** vulnerability of stakes is not taken into account and IRIP hazard levels 4 and 5 are considered at risk and **(b)** and **(c)** vulnerability of stakes is taken into account in various manners based on the principle in which "the higher the vulnerability, the lower the hazard level triggering risk".

**Table 3.** Evaluation criteria used in the study. In the computation of the $\chi^2$ test, the number of degrees of freedom is 1.

|  | Formula | Interpretation |
| --- | --- | --- |
| Probability of detection (POD) | $\dfrac{(\text{Hits})}{(\text{Hits})+(\text{Misses})}$ | Varies from 0 to 1 <br> Perfect score: 1 |
| False-alarm ratio (FAR) | $\dfrac{(\text{False alarms})}{(\text{Hits})+(\text{False alarms})}$ | Varies from 0 to 1 <br> Perfect score: 0 |
| $\chi^2$ test | $\sum \dfrac{(\text{Obtained-Theoretical})^2}{\text{Theoretical}}$ | $P(\chi^2 \geq 10.83) = 0.001$ <br> "Highly significant" <br> $P(\chi^2 \geq 7.88) = 0.005$ <br> "Very significant" <br> $P(\chi^2 \geq 6.63) = 0.01$ <br> "Significant" |

based on the modified contingency tables of step 4.1 or step 4.2 if the latter is relevant.

After these four steps, the final values of the quantitative performance measures are obtained, characterizing the performance of the IRIP mapping model.

### 2.3 Case study

#### 2.3.1 Presentation of the study area

The case study is the 80 km railway line between Rouen and Le Havre (Fig 4). This railway has been operating since 1847. It is a strong stake for the region, as it connects Paris to Le Havre within about 2 h and connects Paris to the major fluvial and sea ports of Rouen and Le Havre. It is located in Pays de Caux, an area known for being affected by intense surface runoff (e.g. Cerdan et al., 2002; Martin et al., 2010). The land use is mainly agricultural. Soils, composed of silts and clays, are sensitive to slacking (formation of a crust lowering the infiltration capacity significantly; Cerdan et al., 2002). The catchment intercepted by the railway has a total area of about 500 km$^2$. Only two streams cross the railway, but the dry thalweg network that can be activated during a rainfall event is very dense (Fig. 4).

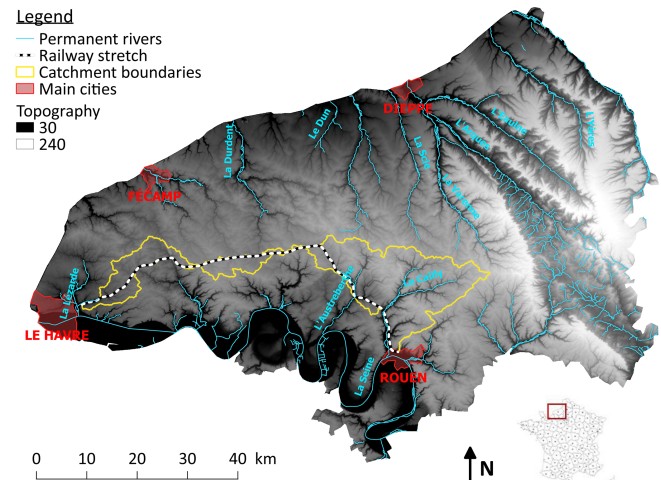

**Figure 4.** TS3 Map of the study area in Normandy (northern France). The yellow contour is the boundary of the catchments intercepted by the Rouen–Le Havre railway (line in black and white). Blue lines are the permanent river courses. One can note the dense network of dry thalweg (darker on the DEM) upstream of rivers that can be activated during a rainfall event.

**www.nat-hazards-earth-syst-sci.net/20/1/2020/** Nat. Hazards Earth Syst. Sci., 20, 1–20, 2020

### 2.3.2 Application of the IRIP model

Three input maps were used to produce the IRIP maps. The retained GIS layers are easily available and allow testing the IRIP model with standard data. The topography was described using the IGN BD ALTI © Digital Terrain Model (DTM) with a 25 m raster resolution. Land use was described using a 1/2500 land use map (https://mos.normandie.fr/TS4) of the Upper Normandy region from 2009. Pedology was taken from the European Soil Database (ESDB) v2.0 with 500 m resolution. The IRIP model parameterization used in this study is presented in Table 1. Given the little local knowledge of the study area, the thresholds defining the classes as favourable or not favourable to runoff were computed using the classification method proposed by default in the IRIP model, contrary to the application by Lagadec et al. (2018) that used values derived from local expertise. Thresholds for the topographic index and slope indicators were therefore defined using the classification method. The threshold of the drained area indicator was fixed to 2.5 ha following sensitivity tests performed by Lagadec (2017) to identify, in this specific catchment, a minimum surface from which significant surface runoff can be generated. Other thresholds for soil depth, hydraulic conductivity, slacking, erodibility and upslope area sensitive to runoff generation were chosen as in Lagadec et al. (2018).

### 2.3.3 The database of impacts on the railway

For its internal needs in terms of risk management, SNCF, the French railway company, has set up a quite systematic archiving system of all the incidents and disruptions of the train traffic as well as of all the works carried out on the railway tracks. This information is archived either in digital databases, available since the 1990s, or in paper format. Paper archives are located in several places in France according to the date the documents were produced. For the present study, the objective was to gather, on the Rouen–Le Havre railway, all the registered impacts related to runoff and the associated information, from the creation of the line to today.

The database was created in two steps: a data collection step and a data processing step. The collection of impact data was carried out with the support of archival expertise. Four archive sites were visited, depending on the age of the documents. It was necessary to define a limited amount of information that had to be collected and that was relevant to runoff. This is provided in Table 4, which describes the two tables that were filled by the archivist when consulting the archives and that were relevant for the next phase of data processing. The first table describes the source documents of interest, and the second table describes the runoff-related events. Archives are organized according to railway kilometric points (KPs) and earthworks. The description of the location of runoff-related impacts makes reference to the KPs and

earthworks, so this information was retained in the event description (Table 4). Indeed, earthworks are relevant elements for dividing the railway tracks into meaningful sections, with regards to its hydraulic operation. Earthworks are designed to insert the railway track into its environment while respecting technical constraints such as a maximum allowed slope to ensure electric traction and braking in good conditions. Earthworks modify the natural surface topography and therefore water flow paths. Four types of earthworks can be distinguished: embankment to cross thalwegs or valleys, excavations to follow longitudinally valleys or to cross small ridges, and mixed profiles to cross hillslopes and quasi-flat profiles (see Fig. 5 railway profiles). Note that a digital GIS layer describing the railway tracks and the location of the various earthworks and their characteristics was created for this study. The challenge in data collection was to manage the diversity of formats (paper, digital) and to manage duplicates. The document collection work took 4 months; 506 documents were retained and inventoried, dating from 1903 to 2017.

The data processing step consisted of retracing the history of each impacted area, the circumstances of the incidents and the work undertaken up to the current situation. One difficulty was to manage the uncertainties, particularly for the location of impacts, and to determine whether they were really direct consequences of an intense runoff event. Finally, the database consists of 59 sections impacted at least once, ranging from point zones to a 1.3 km long section. All the sections impacted at least once represent a cumulative length of 12 km over the 80 km of the railway, or 15 % of the length studied. A geographic information layer of georeferenced impacts with their date, type and uncertainty was finally created, allowing its overlay with the IRIP maps.

There are two assumptions behind the use of this database for the evaluation of the IRIP maps. First, the duration of the period over which impacts are recorded (about 1 century) is long enough that each section may have experienced a possible damaging event. Consequently, the database can be assumed comprehensive. Second, it is assumed that land use types have not changed drastically in this area dominated by agricultural land.

### 2.4 Application of the evaluation methodology to the case study

*Step 1*. The database of runoff-related impacts covers more than 1 century. We can thus assume that all the catchments intercepted by the railway may have experienced a runoff event. We could therefore consider all the catchments intercepted by the railway to be the evaluation area. However, recorded impacts are only located on the railway track. Thus, the evaluation area must be restricted to the railway track itself. However, to account for uncertainty on the location of impacts and of the DTM inaccuracy, a 25 m buffer area was considered on both sides along the railway line. Other values

**Table 4.** Information collected on the documents collected in the archives on runoff-related impacts **(a)** and on the events related to runoff **(b)**.

| **(a)** Information on documents | **(b)** Information on runoff-related events |
| --- | --- |
| Document number (unique identifier) | Event number (unique identifier) |
| Location of the document (where the document is archived) | Start kilometric point (KP) |
| Code of the document | End kilometric point (KP) |
| Date of the document | Name of the earthwork |
| Typology of the document (correspondence, report, etc.) | Date |
| Link to the numerical copy | Type (incident, works, observation, etc.) |
| Remarks | Remarks |

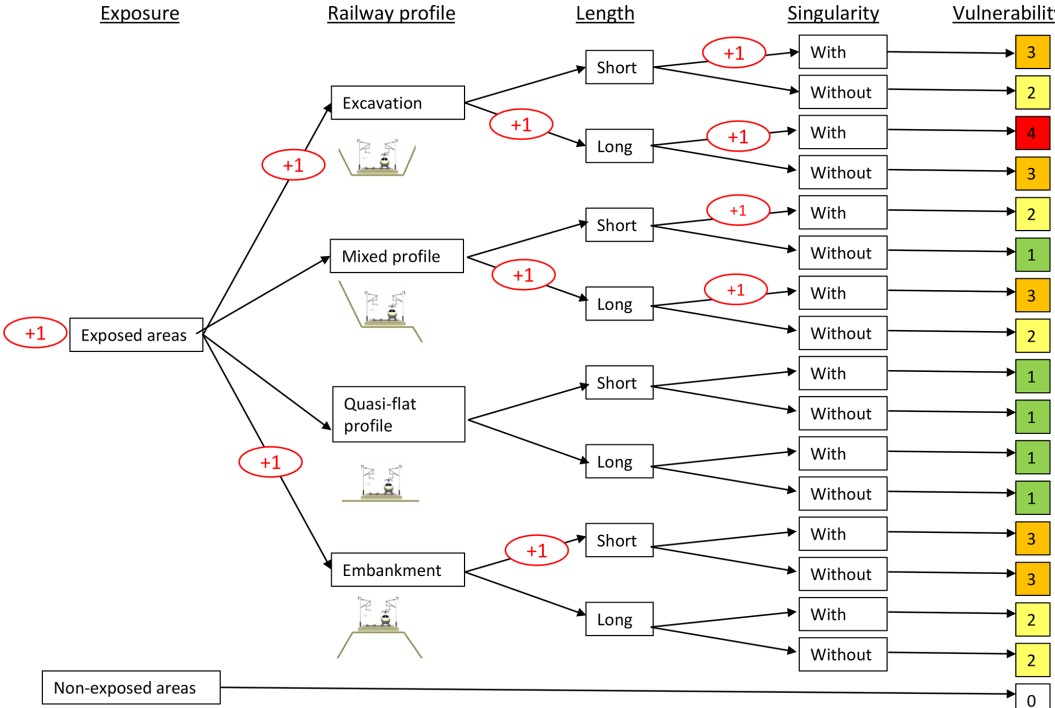

**Figure 5.** Vulnerability tree of the railway sections (also called earthworks) based on expert judgement. Each column corresponds to one criterion considered when computing the vulnerability of the section: the exposure (non-exposed sections are long tunnels or viaducts; column 1), the type of railway profile (column 2), the length of the section (column 3) and the existence of a singularity (level crossings, road bridges, or tunnel inlets or outlets; column 4). The +1 in the red circles indicates that 1 is added to the vulnerability score of the section to provide the final score that appears in the last column of the figure.

of the buffer were tested in Lagadec (2017), but a 25 m buffer was considered to be the most relevant value, given the 25 m resolution of the DTM. We also considered both sides, up and down slopes, as both progressive and regressive erosions were observed around the track.

*Step 2.* The railway track vulnerability was defined using a decision tree, where scores were assigned to each branch of the tree (Fig. 5). The decision tree was built from expert judgement and verified using the impact data (Lagadec, 2017). Four criteria were considered to build the decision tree for the 182 sections of earthworks that were used to divide the railway line into meaningful sections. Earthworks were divided into four types: embankment, excavation, mixed pro-

file and quasi-flat profile. Furthermore, this segmentation of the railway is consistent with impact recording that was assigned to an earthwork type (see Sect. 2.3.3). The four criteria considered to compute the vulnerability scores are as follows.

– *(1) The exposure.* Unexposed areas are sections of long tunnels or large viaducts. The other sections are considered to be exposed and get a vulnerability score of 1.

– *(2) The type of profile and (3) its length.* The types of profiles considered the most vulnerable are long excavations and short embankments. Long excavations are prone to flooding, they have to handle more water from

the slopes they intersect and they are more likely to experience malfunction of the drainage structures. Short embankments are suitable for loading by runoff. They can play the role of a hydraulic barrier. The median earthwork length of the sample was used to separate short and long earthworks.

- *(4) The presence of a singularity*. The singularities are level crossings, road bridges, or tunnel inlets or outlets. These are areas likely to experience arrival of water on the platform. These singularities generally constitute discontinuities in the topography of the work. If one of these singularities is present in the envelope of an earthwork, its vulnerability score is increased by 1.

For defining the hazard, two IRIP maps were considered: the susceptibility maps for transfer and for accumulation that can be related to erosion and flooding, respectively (Lagadec et al., 2016b). Both maps were combined into a unique map that is the union of both maps; i.e. each pixel retains the maximum level of both maps. A value of the hazard level was assigned to each of the 182 sections of the railway track where a vulnerability score was also assigned. This value corresponds to the maximum value of susceptibility to transfer or to accumulation in the 25 m buffer zone on both sides of the railway section. Vulnerability and hazard were combined following Fig. 3c, where red boxes are considered at risk. Performance indicators were also computed for the combination of hazard and vulnerability illustrated in Fig. 3a (where vulnerability of the railway track is not taken into account and levels 4 and 5 are considered at risk) in order to illustrate the impact of taking vulnerability into account in the evaluation method.

*Step 3*. Performance criteria of the IRIP model were computed using the measures defined in Sect. 2.2.3.

*Step 4*. Mitigation measures were considered in a second step. Structural and non-structural mitigation measures were considered and inventoried along the whole railway. Structural measures include all the hydraulic structures (drainage structures along or below the railway track, retention ponds, etc.) that were built to help water flow circulation. At SNCF, non-structural measures include surveillance patrols in case of bad weather. These patrols target, as a priority, the sections registered in what is called the bad-weather tours. These are defined using local knowledge on the hazard exposure or on the specific infrastructure vulnerabilities. They provide increased and targeted monitoring in case of bad weather and early response if needed.

We modified the computation of the performance criteria by taking into account mitigation measures (presence of a hydraulic structure in one section or section registered in the bad-weather tour) as follows. In step 4.1, we moved the sections where a mitigation measure is present but no impact was recorded from false alarm to hit. In step 4.2, we moved the sections where a mitigation measure is present but no impact was recorded from false alarm to hit *and* moved the sec-

tions where a mitigation measure is present and the section was not tagged at risk by the IRIP model from correct negative to miss. In order to quantify the impact of each mitigation measure on the evaluation criteria, they were recomputed with the following methods:

- by taking into account hydraulic works only,

- by taking into account bad-weather tours only,

- by taking into account both hydraulic works and bad-weather tours.

After step 4, the final evaluation criteria of the IRIP model were obtained.

## 3   Results

### 3.1   The IRIP maps

Figure 6 presents the IRIP susceptibility maps for the generation, transfer and accumulation of runoff. The map of susceptibility to runoff generation shows a high sensitivity to the genesis of runoff on agricultural plateaus, an even larger one in urbanized areas and a lower one on slopes that are more vegetated. The IRIP map of susceptibility to runoff transfer shows strong potential for erosion to produce mudflows and mass flows along the uphill slopes of the main thalwegs. Other small thalwegs with a high susceptibility level to runoff transfer are also scattered throughout the area, showing areas potentially sensitive to erosion. The map of susceptibility to runoff accumulation highlights all the preferential flow paths that have a high value of the susceptibility level. This includes not only the perennial streams but also dry thalwegs, the latter being located in headwater catchments with flatter areas.

### 3.2   Results of the evaluation method

Using Fig. 3c to combine hazard and vulnerability, Fig. 7 provides illustrations of the application of the evaluation method and of the building of the contingency tables.

In the Fig. 7a (Area A), for the two "H" (see caption in Fig. 7) sections, the vulnerability scores are greater than or equal to 2, and there are pixels with IRIP levels greater than or equal to 4; these sections have already been impacted at least once, and they are therefore hits. The "CN" section has a vulnerability of 2, its maximum IRIP level is 3 and there has been no impact, so the section is a correct negative. The section "FA" has a maximum vulnerability score of 4, so the level IRIP 3 is sufficient to consider this section at risk; nevertheless no impact was observed here. The section is thus counted as a false alarm.

Figure 7b (Area B) illustrates how mitigation measures are taken into account in the analysis. The H section has a vulnerability score of 2 and a maximum IRIP level of 5, so it is

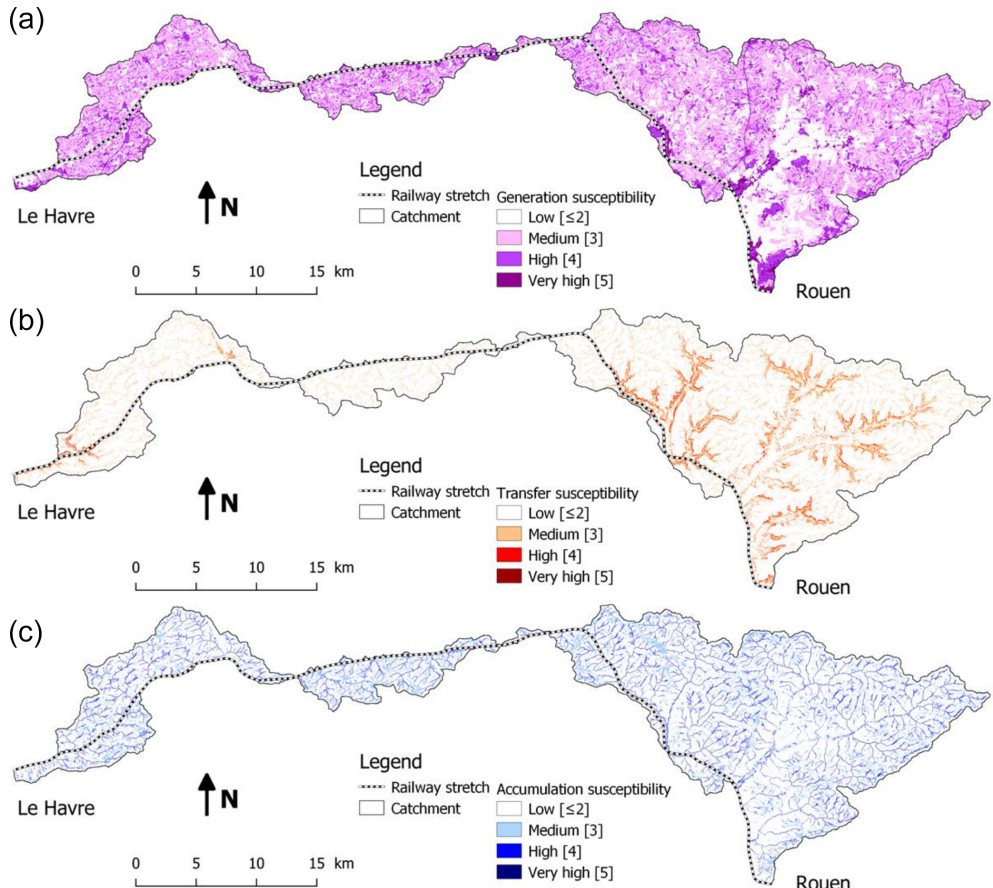

**Figure 6.** IRIP susceptibility maps to runoff generation **(a)**, transfer **(b)** and accumulation **(c)**.

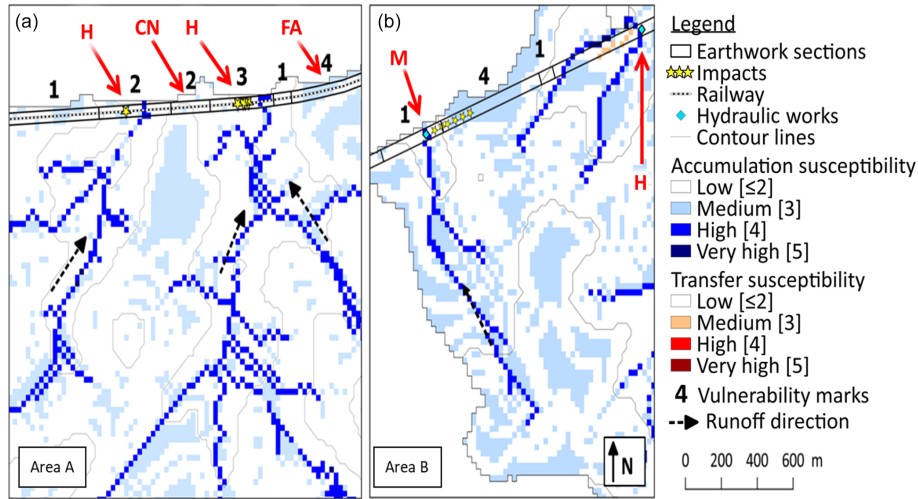

**Figure 7.** Illustration of the evaluation area and of the building of contingency tables on two sub-areas: **(a)** Area A and **(b)** Area B. The evaluation area is the buffer (black contours) along the railway line (black dotted line). This is divided into sections (earthwork sections) to which a vulnerability score is assigned (black numbers in the figure). Impacts are the yellow stars. Hydraulic infrastructures are marked with blue diamonds. IRIP susceptibility levels appear in red for transfer and blue for accumulation (only levels 3, 4 and 5 are drawn in the figure). Red arrows show the value assigned to the railway sections in the contingency table, where H means hit, CN means correct negative, M means miss and FA means false alarm.

considered at risk, yet no impact has been reported. This section should be assigned to the false-alarm class, but we note that it is equipped with a crossing work under the railway that could play a role in protection against the hazard. The location of this infrastructure shows that the hazard is indeed present at this location and that the IRIP map is correct, so the section is finally rated as a hit once mitigation measures are taken into account. The M section has a vulnerability of 1, which requires a IRIP level of 5 to be considered at risk, but the maximum IRIP level is 4, so the section is not considered at risk. However, as an impact occurred, the section is considered to be a miss. Note also that, even if no impact had been recorded, the section would have been moved to a miss according to step 4.2 of the methodology, as a mitigation structure is present but the IRIP model does not classify the section as being at risk.

The results of the evaluation along the whole railway line are reported in Table 5. Five results are presented: column (1) – when neither the vulnerability nor the mitigation measures are taken into account (meaning that hazard and vulnerability are combined following Fig. 3a); column (2) – when only the vulnerability is taken into account following Fig. 3c; and columns (3) to (5) – when the vulnerability (Fig. 3c) and the mitigation measures are taken into account, including, respectively, hydraulic works (column 3), bad-weather tours (column 4), and both hydraulic works and bad-weather tours (column 5). In columns (3) to (5), performance criteria are given for step 4.2, and the values in brackets correspond to step 4.1.

The results show that the POD increases from 86 % (column 1) to 93 % (column 2) and the FAR decreases from 62 % to 58 % when vulnerability is taken into account in computing them. The results in terms of $\chi^2$ present the same trend, with the significance increasing when vulnerability is considered. It can be noticed that from column (1) to column (2), four missed impacts are moved into hit. These are impacts that occurred on highly vulnerable railway sections that can experience damage with a very low hazard exposure level. Likewise, six false alarms are moved into correct negatives because the railway sections were not enough vulnerable.

The results of Table 5 (columns 3 to 5) show that the impact of taking into account mitigation measures on the performance criteria is large. When considering step 4.1 only (figures in brackets), the number of misses remains the same as in columns (2) but the number of false alarms dramatically decreases. This leads to a similar POD but a significant decrease in FAR. When considering step 4.2, i.e. assuming that the existence of mitigation measures proves the existence of a risk, the number of false alarms is the same as for step 4.1, but the number of misses increases when compared to column (2), the larger value being obtained when both hydraulic works and bad-weather tours are taken into account. When compared to step 4.1, the FAR remains the same, but the POD decreases compared to column (2). Nevertheless, POD values remain similar to the one obtained when no vulnerability is taken into account (column 1). The values of $\chi^2$ show that the relation between risk and impacts is significant in column (1) when vulnerability is not taken into account and highly significant when vulnerability and mitigation measures are taken into account. The results in columns (2) to (5) present very encouraging values, highlighting the added value of the IRIP maps, and of the vulnerability and mitigation measure characterization, for the evaluation of the IRIP model.

## 4 Discussion

### 4.1 Relevance and limitations of the evaluation method

The results of the evaluation method, as applied to the IRIP maps, show the interest of considering both the vulnerability of the railway and mitigation measures taken to lower risk in the computation of performance criteria. These two factors are essential for making an accurate and fair comparison between the localized runoff-related impact data and the IRIP maps. When both are not considered, POD values are high, but the rate of false alarms is also very high. Note that the term false alarm does not mean that the information is false but that it cannot be proven: even if no past impact occurred, an impact could possibly happen in the future.

When vulnerability of the railway is taken into account, both the POD and FAR are improved (comparison of columns 1 and 2 in Table 5). When evaluating the performance of a road cutting warning system, Versini et al. (2010a, b) and Naulin et al. (2013) also showed that it was essential to incorporating road vulnerability in their computation of POD and FAR to get meaningful results. However, the results of the performance criteria depend on the choice made to assign hazard to the section (here we chose to use the maximum hazard – as discussed in Sect. 2.2.2 – value within the section) and on the choice made to combine hazard and vulnerability (i.e. Fig. 3c). Lagadec (2017) compared combinations shown in Fig. 3b and c but found similar performance criteria. The difference between performance criteria when vulnerability is not taken into account (column 1 in Table 5) or is taken into account (columns 2 in Table 5) shows that considering the vulnerability of the railway has a much higher impact on the performance criteria values than the way hazard and vulnerability are combined.

Lagadec et al. (2018) tested another way to assign the hazard value to one section by declaring that a section was at risk if the percentage of the areas with values larger than a susceptibility threshold (four in Lagadec et al., 2018) was higher than a percentage threshold (10 % in Lagadec et al., 2018). In this case, the user must choose two thresholds, the values of which will strongly affect the evaluation measures values. When defining the hazard level using the maximum susceptibility value in one section, as used in this study, these subjective choices are avoided. However, when runoff-related im-

**Table 5.** Performance criteria assessing the predictive power of the IRIP model in identifying sections with a proven risk using different methods to take into account vulnerability and mitigation measures: without taking vulnerability into account according to Fig. 3a (column 1), when taking vulnerability into account according to Fig. 3c but not mitigation measures (column 2), and when taking vulnerability into account according to Fig. 3c and mitigation measures (columns 3 to 5). In columns (3) to (5), the figures correspond to step 4.2, and those in brackets correspond to step 4.1 of the methodology.

| | (1) Without taking vulnerability into account (cf. Fig. 3a) | (2) When taking vulnerability into account (Fig. 3c) but not mitigation measures | (3) When taking vulnerability (Fig. 3c) and hydraulic works into account | (4) When taking vulnerability (Fig. 3c) and bad weather tours into account | (5) When taking vulnerability (Fig. 3c) hydraulic works and bad-weather tours into account |
|---|---|---|---|---|---|
| Number of hits | 51 | 55 | 85 (85) | 67 (67) | 95 (95) |
| Number of false alarms | 83 | 77 | 47 (47) | 65 (65) | 37 (37) |
| Number of correct negatives | 40 | 46 | 43 (46) | 37 (46) | 35 (46) |
| Number of misses | 8 | 4 | 7 (4) | 13 (4) | 15 (4) |
| Probability of detection: POD (%) | 86 | 93 | 92 (96) | 84 (94) | 86 (96) |
| False-alarm ratio: FAR (%) | 62 | 58 | 36 (36) | 49 (49) | 28 (28) |
| $\chi^2$ | 7 | 19 | 36.8 (46.1) | 9 (27.9) | 26.7 (59.8) |

pact data are available and if IRIP hazard is to be used for operational purposes, adjustment of the method to assign IRIP hazard may be necessary so that areas tagged as at risk are meaningful for the territory managers and in order to prioritize areas requiring protection measures.

There are, however, limitations of the vulnerability–hazard combination, as illustrated in Fig. 7b (Area B). The section M is rated not very vulnerable, with a score of 1. Thus level 5 is required in the IRIP maps to consider this zone at risk. This is not the case, as the maximum level of the IRIP maps is 4. However, this area has already been impacted by intense runoff as impacts were recorded, and it is also equipped with a hydraulic structure crossing under the track. This section is therefore at risk but classified as missed impact. We can see that the IRIP map shows a specific arrival of runoff, so the map looks correct. But the section vulnerability score is only 1, leading to consider it not at risk. Therefore, the vulnerability classification is obviously deficient in this example and should be modified to better take into account the specificities of this type of configuration (quasi-flat profile). Further discussion with railway experts could lead to increasing the vulnerability score of quasi-flat profiles.

The results presented in Table 5 also highlight the large impact of the way mitigation measures are taken into account (step 4 of the methodology) on the final values of the performance criteria. Considering only step 4.1 (moving the sections where a mitigation measure is present but no impact was recorded from false alarm to hit) slightly increases POD values and dramatically decreases FAR values (see values in brackets in columns 3 to 5 in Table 5). The decrease in FAR is larger when bad-weather tours only are accounted for than when hydraulic structures only are taken into account. The lowest FAR values are obtained when considering both mitigation measures. Considering also step 4.2 (additionally moving from correct negative to miss for the sections where a mitigation measure is present but the section was not tagged at risk by the IRIP model) leads to a decrease in POD but

does not change the FAR when compared to step 4.1. Applying step 4.2 implicitly means giving the same status of proven risk to mitigation measures and to localized observed impact. Thus, it could be possible to directly use the presence of a hydraulic work or a bad-weather tour as explaining factors of the observed localized impacts and to compute the corresponding contingency tables. The results are presented in Table 6, where the performance criteria were computed using hydraulic works only (Table 6, column 1), bad-weather tours only (Table 6, column 2) and a combination of both sources of information (hydraulic works *or* bad-weather tours; Table 6, column 3) for explaining the recorded runoff-related impacts. The results show that the hypothesis that hydraulic works and runoff-related risk are independent cannot be rejected ($\chi^2$ not significant); i.e. hydraulic works have low predictive power with respect to the occurrence of a risk. On the other hand, bad-weather tours have predictive power but lower values of POD (49 %) than the IRIP model without mitigation measures (93 %; comparison of column 2 in Table 6 and column 2 in Table 5). The FAR value is lower than for the IRIP model without mitigation measures (42 % when compared to 58 %) but is not so different. Finally, when the presence of hydraulic works and that of bad-weather tours are combined (Table 6, column 3), the POD increases to 68 % when compared to considering hydraulic works only or bad-weather tours only. The FAR is 56 %, an intermediate value between the one of hydraulic works only (60 %) and bad-weather tours only (42 %). In any case, the predictive power of the IRIP model is higher than when considering hydraulic works or bad-weather tours to be a proxy for the risk of intense runoff.

This shows that the use of mitigation measures as proxy data must be done with caution. The correct use depends on the accuracy of the data and the degree to which they are related to proven risk. In the context of the railway case study, the following elements must be taken into account. The construction of hydraulic works or the design of tours takes into

**Table 6.** Performance criteria assessing the predictive power of the presence of hydraulic works or bad-weather tours in identifying sections with proven risk.

|  | (1) Impacts explained by hydraulic works | (2) Impacts explained by bad-weather tours | (3) Impacts explained by hydraulic works or bad-weather tours |
|---|---|---|---|
| Number of hits | 22 | 29 | 40 |
| Number of false alarms | 33 | 21 | 51 |
| Number of correct negatives | 90 | 102 | 72 |
| Number of misses | 37 | 30 | 19 |
| Probability of detection: POD (%) | 37 | 49 | 68 |
| False-alarm ratio: FAR (%) | 60 | 42 | 56 |
| $\chi^2$ | 2 (not significant) | 20.6 | 11.1 |

account not only the hazard parameter but also the vulnerability and the criticality of the stake. A bad-weather tour is preferably designed on a section that is critical regarding the train traffic management or on sections with known structural weaknesses. Bad-weather tours are not precise; they often involve long linear areas of the railway, and all sections of the tours may not be relevant to runoff risk. This is why we used them in step 4 to refine the computation of performance criteria, in particular as explanatory factors of false alarms and not as elements proving the existence of a risk. The results of the evaluation criteria obtained after step 4.2 assume that the existence of mitigation measures means proven risk, which is not the case. Therefore, they provide the most pessimistic POD values for the evaluation of the IRIP maps. The results presented in Table 6 also highlight that the bad-weather tour is more reliable proxy data for runoff-related risk than hydraulic works. But this may be due to the number of railway sections concerned in one bad-weather tour, whereas local mitigation measures only affect one section.

One of the reasons for this low predictive power of hydraulic works could be the following. Blockage of culverts or drainage pipes is a common problem in the railway context. In addition to blockage related to a particular intense event, progressive filling of the infrastructure by diffuse sediment transport is also a difficulty, since there are a large number of small hydraulic works that are difficult to maintain. Unfortunately, the information on blockage of hydraulic works is rarely documented in the reports about the impacts found in the archives. On the other hand, other hydraulic works are well dimensioned and very efficient. Thus, hydraulic works can sometimes increase the vulnerability and sometimes decrease it. Therefore, it was not possible to consider this information to be a reliable source of information for proven risk in the evaluation methodology or in the vulnerability tree. On the other hand, the IRIP map of susceptibility to transfer, by highlighting areas prone to sediment transport, can allow management and warning to be concentrated on these areas.

Another limitation of the evaluation presented in the paper is related to the runoff-related impact database itself. As mentioned before, the location of impacts is sometimes not

very accurate and may alter the computation of the performance measures. Furthermore, although it covers more than 1 century of data, the database may not be comprehensive, which could affect the false-alarm ratio if all the occurred impacts have not been recorded. Moreover, the evaluation was conducted assuming (see Sect. 2.3.3) that each section of the railway had the opportunity to be affected by runoff, i.e. that each section of the railway had the opportunity to be affected by an intense rainfall event. If it were not the case, the IRIP model could indicate a risk in a section that would not have been impacted in the absence of any intense rainfall event at that location. To assess the validity of this working hypothesis, in which "each section had the opportunity to be affected by an intense runoff event", we can calculate the probability of not having experienced a rainfall event of a given return period during 1 century. This probability is less than 0.001 % for a 10-year return period $((1/10)^{100})$, less than 1 % $((1/20)^{100})$ for a 20-year return period, and 13 % $((1/50)^{100})$ CEI for a 50-year return period. Therefore, it can be assumed that each section of the railway had the opportunity to experience a rare event at least once during the data collection period. This shows that, if the database is long enough and of course comprehensive (i.e. all the occurred runoff-related impacts were properly reported), the working hypothesis can be accepted, and, therefore, performance measures can be considered to be not biased. In the present case, the comprehensiveness of the database is exceptional but far from perfect. However, it was the best that could be collected, and the duration of data collection (more than 1 century) ensures that the chosen case study was relevant for assessing the accuracy of the proposed evaluation methodology.

Another point that must be considered is the assumption of a constant land use map for the IRIP map building. It is clear that land use has changed over a whole century, with the development of intensive agriculture and urbanization. Indeed large field crops have replaced the mosaic of small fields crops with hedgerows (the so-called bocage) since the second world war. The IRIP model considers that urban and crop lands are both favourable to intense runoff generation.

In the context of the largest cities of Rouen and Le Havre that are located at the start and end of the railway line, urban growth has no major effect. Loss of grassland and forest is more sensitive. As the IRIP maps were established with the 2009 land use, change in land use over the last century would lead to a possible overestimation of false alarms, as current land use is more prone to runoff than in the past, when the bocage was protecting the land surface from runoff. Land use change could also explain the increasing number of impacts in the recent decade. However, this increase could also be explained by a more comprehensive record of impact statements in SNCF practices during this period.

## 4.2 Impact of the uncertainty and resolution of the IRIP maps

Regarding the influence of the resolution of the input maps on the final maps, several resolutions and qualities of DTMs were compared in Lagadec (2017), with five DTMs ranging from 250 to 5 m on only 30 km of railway for which an accurate lidar DTM at 5 m resolution was available. It showed that there is spatial persistence of information from higher resolutions to coarser resolutions. The analysis also showed that the data acquired by lidar provide very relevant information that helps to understand the phenomenon of runoff. In particular, lidar data provide improved representation of runoff pathways, as they explicitly include linear features such as ditches or roads that are not seen by coarse-resolution DTM but are detected with high-resolution ones. Although it would be recommended to have similar resolutions for the three input maps, accuracy of the DTM is essential for the IRIP model application, as the DTM is used to compute three factors in the transfer susceptibility map and four factors in the accumulation susceptibility map. On the other hand, three factors out of five use the soil map in the building of the susceptibility to runoff generation map. The quality of these data is therefore essential for the interpretation of the susceptibility to runoff generation map. Efforts spent on collecting accurate input data depend on the use of the final maps. Input data resolution also depends on the size of the study area and must be chosen to facilitate map reading and to optimize computing resources. For a large study area, it is recommended to zoom in through successive applications of the IRIP model: identify the most exposed areas with coarser resolutions first and then zoom in with higher resolutions.

When considering coarser resolutions, it becomes difficult to apply the evaluation method proposed in this paper (mainly 75 and 250 m resolution DTMs), as the evaluation zone must be enlarged to account for the larger pixel resolution. The size of the railway sections becomes small when compared to the pixel resolution, so it becomes more difficult to overlay point impacts and IRIP maps pixels. In the same way, high-resolution maps imply adjusting some choices made for the evaluation process, such as the size of the buffer area on both sides of the railway. The way in which

a hazard level is assigned to a section should also be reconsidered, as the chance to get one pixel with a high hazard level is larger if the resolution is higher, and it may not be relevant anymore to mark the whole section as at risk if there is only one pixel with a high hazard level. Computing a percentage of the section with a high hazard level may be more relevant in this case. For these reasons, quantitative evaluation has not been tested yet with high-resolution maps. Only qualitative analyses are provided in Lagadec (2017).

Results of the evaluation method also depend on the values of the parameters chosen to compute the three susceptibility maps (as specified in Table 1). The IRIP model can be applied everywhere without prior knowledge of runoff over the area. However, the relevance of the maps improves significantly when some parameters are adjusted using local knowledge on the area. Examples of such adjustments can be the following.

The drained-area threshold depends on the level of detail expected at the head of the basin, but it remains between 0.5 and 5 ha. Above 5 ha too much information is lost (Lagadec, 2017). Note also that localized impacts of runoff were recorded for catchments of a few hectares, and this consideration also guided the choice of the threshold value.

The break of the slope threshold depends on the calculation method used to compute the break of the slope factor, as the user can choose the number of pixels over which the factor is computed. The number is always an odd number, and three pixels are the minimum number of pixels that must be used. If the number of pixels increases, information on micro-topography becomes less accurate. The number of pixels must be adapted according to the resolution of the DTM: for instance, 3 to 7 pixels are recommended for a DTM of 25 m resolution, and between 9 and 25 are recommended for a DTM of 5 m resolution.

The types of land use that are considered favourable to runoff can also be modified, for example according to the agriculture cycle on a same plot. The modification of these parameters and the evaluation of their relevance depend on the expert conducting the study, CE2 their knowledge of the area and also their objective (precise study or large mesh for larger territories).

There are limitations related to the IRIP model itself, the main one being that the model does not provide quantitative estimates of runoff. The other limitation is that the produced susceptibility maps are relative to the study area, as the thresholds that divide the factors maps into areas sensitive or not sensitive to runoff are computed for the study area. Therefore, it is not possible to compare maps from two areas, and if the study area changes a little, the map will also change a little. There are also limitations related to the application of the model. The IRIP maps of transfer or accumulation strongly depend on the DEM quality, since three and four indicators out of five are derived from the topography. The required computing time is large when large study areas are considered or if the DTM resolution is high. Finally,

there are limitations in the evaluation itself, as the maps of susceptibility to runoff generation were poorly evaluated due to the lack of appropriate data.

The data set used in this study is very comprehensive and includes the three pieces of information required for the application of the evaluation methodology described in Sect. 2. The quality of the data set allowed its use for testing improvements of the IRIP model as done by Lagadec (2017), who used the quantitative measures to test alternative indicators for the building of IRIP maps. This led to recommendations to improve the method, as proposed by Lagadec et al. (2018) and used in the present study. One example is the 2.5 ha threshold value of the drained area chosen to separate the conditions favourable and not favourable to runoff transfer and accumulation (see Fig. 1 and Table 1). To choose this value, Lagadec (2017) performed six simulations, with a drained-area threshold ranging from 0.5 to 100 ha. A threshold value between 1 and 5 ha was a good compromise between performance in explaining impacts and the visual aspect of the maps.

Up to now, all five factors involved in the IRIP maps are given the same weights. The evaluation methodology could be also used to compare non-equal weights in the building of the maps. Methods such as the one proposed by Neuhäuser et al. (2012) could be used for this purpose.

## 4.3 Genericity of the evaluation method

The evaluation method presented in this paper was applied using proxy data of runoff-related impacts on the railway. In this case, the evaluation area was defined as a buffer zone along the railway to account for inaccuracy in the impact location and DTM. Apart from the compilation of a database of impacts on the railway, the combination of hazard and vulnerability required a complete analysis of the vulnerability of the railway and of its characteristics and an inventory of all the hydraulic structures set up to limit impacts related to runoff. This was a huge effort, as all the corresponding information was not digitized yet, but such an effort was very valuable and can be used for other studies. As more and more companies or administrations are setting up databases of the infrastructures they are controlling, such databases are becoming more and more common. We have seen that information on vulnerability and mitigation measures was necessary to decrease the false-alarm ratio and that the impact on the probability of detection depended on the way mitigation measures were taken into account. In any case, without information on vulnerability, but if a compilation of impact data is available, it is possible to compute reliable estimates of the POD but not of the FAR.

In this paper, the evaluation methodology was applied to the railway context, with proxy data related to runoff-related impacts on the railway. This led to a very specific definition of the evaluation area that was restricted to a buffer zone on both sides of the railway. The approach can easily be extended to road networks or any other linear stakes.

Other applications of the evaluation method are in progress. They show that the method can be applied to other types of localized impact data. Two contexts can be distinguished.

- *The availability of a long-term database of impacts over a territory*. This may include impacts on the protected forested domain managed by the French ONF agency (National Forests Office; see also Defrance et al., 2014). This database contains information on damage and protection infrastructure against landslides, gullying and flooding in the Alps and the Pyrenees. It was launched in the 1980s but also contains information on historical events. Given the duration of data collection, an assumption that the whole surveyed territory may have been impacted can be made, and the evaluation area can be defined as the whole forested area covered by the survey. Using these data, the probability of detection can be computed with a good degree of confidence, provided that impact localization is accurate enough, and especially if information on protection infrastructures can be incorporated into the analysis. It is very difficult to get information on vulnerability of the territory, and this information should be defined with local stakeholders.

- *The availability of impact data for a given hydrometeorological event*. In this case, information on rainfall is necessary to define the evaluation area, as no impact will be observed if no rainfall or only low-intensity rainfall was recorded. The evaluation area can be defined using rainfall data and a given rainfall intensity threshold. Given that the events are often much localized, the use of radar rainfall data with a short time step (e.g. 5 min time step) is recommended, as shown by Marra et al. (2016) for landslides. The rainfall threshold triggering localized impacts can be assessed if a time series of spatialized rain and georeferenced localized impacts are available for the same storm event. The principle consists of searching for the maximum rainfall for the different impacts over different durations from 5 min to 1 h. The durations of interest are based on the assumption that the higher the average hazard level that is located in the vicinity of the impact, the lower the amount of rain required to trigger the impact. Initial results point to a relevant duration of 15 min to 1 h. Once the duration has been selected, the minimum rainfall intensity over this duration is selected and considered to be the rainfall threshold necessary to trigger all observed impacts. This assumption allows restricting the model's evaluation area to the areas where it has rained enough. If data collection on impacts is comprehensive and information on protection infrastructure is available, POD and FAR values can be computed with good accuracy.

The evaluation methodology has been designed for localized impact data. However, the avenue of very high-resolution remote-sensing information opens perspectives for the evaluation of a mapping method, like the IRIP model. Indeed, those techniques are now able to provide accurate information on erosion and gullying (e.g. Desprats et al., 2013; Eckert et al., 2017) that could be used as verification data for the IRIP maps. Such analysis is in progress using data from the satellites Pléiades 1A and 1B, which provide 0.7 m image resolution, for the 15–16 October 2018 rainfall event in the Aude department in France, with 15 fatalities and damage of about EUR 220 million. Even if river flooding is responsible of a large part of damage, runoff outside of the river network was also observed that particularly affected agricultural land. A database of insurance claims related to damage to agricultural land is being collected. It will allow the assessment of the added value of high-resolution satellite images in the evaluation of the IRIP maps when compared to the use of localized impacts. The evaluation method proposed in this paper may, however, need some adaptation to be used with remote-sensing images.

### 4.3.1 Relevance of the IRIP maps for risk assessment and design of mitigation measures in the railway context

The evaluation method presented in this paper and the results obtained with a comprehensive database of runoff-related impacts raise confidence in the relevance of the IRIP maps and of their potential for use in risk management studies and confirm this potential that was highlighted in previous studies in the railway context. Lagadec et al. (2016a) showed qualitatively the usefulness of IRIP maps combined with high-resolution radar images of rainfall for post-event survey after an intense rainfall event that damaged the St-Germain-des-Fossés–Nîmes line in the Gard department in 2014. Lagadec et al. (2018) showed that IRIP maps were consistent with an expert hydraulic diagnostic of the Bréauté–Fécamp line in Normandy in prioritizing railway track rehabilitation works. There are several advantages to using IRIP maps: time can be saved and accuracy in the results increased by using these maps as a new source of information to inform field expertise. The maps help to prepare the expertise and to better understand the context, and once on site, they allow focusing on specific areas and moving to certain points in the catchment area (areas of runoff generation, erosion, deposition or stagnation). For studies on larger linear areas, the method automatically identifies all exposed areas. By crossing with the infrastructure configuration, a pre-diagnosis of the areas at risk is obtained. IRIP is therefore a relevant tool for helping to identify runoff hazard.

For the risk management and technical-solution-definition phases, runoff maps can also provide useful information. The maps represent the hydrological surface processes over the entire watershed around the railway. This can help in the im-

plementation of actions on hillslopes in choosing the location of solutions and in adapting them to hydrological processes according to the zones (erosion, deposits and stagnation). However, working outside of the railway right of way is still difficult today. On the one hand, from a legal point of view, it is necessary to obtain the agreement of the plot owners and to establish contracts for maintenance operations. In addition, such nonstandard technical solutions are often not referenced in quality control procedures, as they rather propose dimensioning (in flow and volume) of complete networks bringing water to an outlet. On the other hand, the implementation of alternative techniques favours a locally adapted solution that can be validated through risk analysis. Nevertheless, acting on the slope is sometimes the only sustainable solution to managing runoff. Sediment inputs are difficult to manage by conventional hydraulic structures, and the lack of space in the railway right of way makes it difficult to implement appropriate solutions. As the constraints of discharges into the environment are very restrictive, managing runoff may require the creation of retention basins, a solution that is often incompatible with the available space. It would be interesting to reduce inflows, for example by creating retention pools in accumulation areas or rehabilitating or creating wetlands (Fressignac et al., 2016), by setting up fascines on the transfer axes to trap sediments and avoid soil losses, by developing grassy stripes on the deposit areas to allow fines and sludge to spread, by avoiding bare land, or by favouring vegetation to increase infiltration capacity on areas sensitive to runoff generation. These soft hydraulic techniques are to be used in addition to the traditional hydraulic techniques used to manage exceptional events. In the long term, the actions on the hillslope limit the degradation of railway infrastructure elements, increase safety and reduce economic losses. The interest is also ecological by creating wet or wooded areas with an improved social perception of railways in the landscape. The runoff problems encountered at a point of the railway infrastructure generally also have an impact upstream of the infrastructure. Whether for urban areas (flood risk) or agricultural areas (erosion risk), runoff also needs to be controlled upstream. It would seem more relevant and technically more efficient to manage runoff in a distributed manner throughout the watershed. There are many obstacles to such control: complex legislation, difficulty of communication, differences in deadlines and budget according to the actors. Having a visual and educational tool, illustrating the downstream impact of an upstream action, and the interconnection of issues on the territory can promote such synergy. The IRIP map can be a tool to assist in such eco-design.

## 5 Conclusions

The paper presents an evaluation method suitable for assessing the relevance of susceptibility maps to intense runoff, using proxy data of localized runoff-related impacts. The

evaluation method takes into account not only the hazard knowledge but also the knowledge of the vulnerability of the study area concerning the considered hazard and of mitigation measures taken to lower the risk. The methodology was validated using a comprehensive database of runoff-related impacts on an 80 km railway in northern France, covering more than 1 century of operation, and applied to the maps produced using the IRIP (indicator of intense pluvial runoff) maps. Due to the quality of the data set, we were able to quantify the impact of taking into account or not taking into account the information on vulnerability and also different methods for accounting for mitigation measures on the computation of performance criteria. Information on vulnerability and mitigation measures can be time-consuming to collect. However, it is essential for obtaining meaningful performance measures characterizing the accuracy of the map, and it is also needed to get a good appraisal of the risk. It would be interesting to gather the same type of information in other climatic, pedologic and land use contexts. The methodology proposed in the paper is generic enough and can be extended to other sources of localized impact data and to other mapping methods of susceptibility to runoff. In order to capitalize on runoff-related impact data acquisition, one perspective is to build a platform where stakeholders could provide their runoff-related impact data and benefit from an online QGIS plugin implementing the IRIP model. This could contribute to increased runoff knowledge and understanding and improving runoff risk management.

*Data availability.* The impact and damage data used in this study are confidential and the property of SNCF Réseau and cannot be made available publicly. The study was performed using the IRIP ©software, property of SNCF, that cannot be made available either. A new version of the IRIP model is, however, under development as an open-source QGIS plugin.

*Author contributions.* Most of the work presented in this study was conducted by LRL during her PhD thesis under the supervision of IB, LM, PB and BC. IB wrote a first English version of the paper that was revised and contributed to by the other authors.

*Competing interests.* The authors declare that they have no conflict of interest.

*Special issue statement.* This article is part of the special issue "Natural hazard impacts on technological systems and infrastructures". It is a result of the EGU General Assembly 2018, Vienna, Austria, 8–13 April 2018.

*Acknowledgements.* We thank Judicaël Dehotin, a key proponent of the development of the IRIP method. Corentin Descours helped in mapping the Rouen–Le Havre railway profile and its vulnerability during its use. Sixtine de Bejarry performed the data collection of impacts and/or damage on the Rouen–Le Havre railway.

*Financial support.* This research has been supported by the Agence Nationale de la Recherche et de la Technologie (ANRT; grant no. CIFRE 2014/0723) and MTES (French Ministry of Ecological and Solidarity Transition).

*Review statement.* This paper was edited by Elena Petrova and reviewed by Axel Bronstert and Diego Cerrai.

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

**Remarks from the language copy-editor**

CE1    Please give an explanation of why this needs to be changed. We have to ask the handling editor for approval. Thanks.

CE2    Please note that "their" does not refer to the plural here. This is the correct conjugation, even if the expert only refers to one person.

**Remarks from the typesetter**

TS1    Please add last access date.

TS2    Please add last access date.

TS3    The figure was updated.

TS4    Please add last access date.

TS5    Where should 2006 be added in the text?

TS6    A last access date is not needed when a DOI is given.