# Peer review of "A method to use proxy data of runoff-related impacts for the evaluation of a model mapping intense storm runoff hazard. Application to the railway context"

_Natural Hazards and Earth System Sciences, 2019_

## Referee Comment (RC1) · Anonymous Referee #1 · 30 Jul 2019

This manuscript presents an innovative model regarding the risk of pluvial local flooding and the possible impacts on a traffic infrastructure, in this case a railroad line. The manuscript mainly presents the model concept and an application to the Rouen – Le Havre railway in NW France.

The contents of the paper is innovative, it is written in a well-structured and mostly clear manner and the research and the results are of high relevance for NHESS. The results of te paper are convincing!

I suggest the publication with a few minor amendments and some additional explanations:

Page 2, line 1-5: I think that the options of using a physically based model are written in a rather pessimistic manner. I think that such a model could be applied with a similar data base and with similar results. However, the big difference would be the required time for setting-up such a model and the required computational time. But it could be run with a rather high temporal resolution instead.  Maybe you could elaborate a bit more on those differences.

Page 3, introducing the IRIP model: Can you talk a bit on the temporal resolution. I understood it is a quasi-static model, i.e. no temporal resolution. This should be mentioned. Furthermore, please explain if (and possibly how) variability of rainfall in space is considered and how one may approach/ guess the critical rainfall intensity thresholds.  I also think that one should mention the question of an appropriate / meaningful spatial resolution here. You discuss this well in the discussion chapter, but you may refer here already to this discussion, because it is essential for model application.

Page 3, line39: Typo ("reduced" instead "reduces")

Page 4, line 13: how to guess the "rainfall large than a specified intensity"

Page 4, step 2.2.2: Can you explain why the runoff susceptibility map is not important here

Page 5, line 36: I cannot see easily the thalweg structure in figure 4, can you improve this visibility?

Page 6, line 31: Figure 5 needs more explanations.

Page 11, line 38: Can you elaborate a bit more, if (and if yes, how) this model could account for individual rain storm events. This would be rather interesting.

Discussion section: Did you gain any information about possible blockage of culverts / drainage pipes under the railroad track during heavy rainstorms? If yes, it would be very interesting to read about this. I have been informed about such incidences in Germany during after flash floods. Those created a big problem, when the street or railroad dams were impounded, overflown, eroded and partly broken. I think tis risk is under-estimated if not neglected at all.

Discussion (or conclusions): can you add a paragraph summing the limitations of the model?

Table 3: Where can one get these estimates for model parameters from? Are these the default estimates? Can you give some reasonable parameter ranges?

Figure 2: somehow difficult to understand

Figure 4: Please improve / extend this figure a bit:
  – Include an inlet, where you show the location of this region within France
  – Names of the river are hardly readable.
  – Dry thalwegs difficult to guess.
  – Rouen and Le Havre urban areas could be shown?

---

## Author Comment (AC1) · 23 Sep 2019

**Answers to Reviewer#1 comments**

We thank Reveiwer#1 for his/her comments. The reviewer comments appear in black and the answers appear in blue.

1/ This manuscript presents an innovative model regarding the risk of pluvial local flooding and the possible impacts on a traffic infrastructure, in this case a railroad line. The manuscript mainly presents the model concept and an application to the Rouen – Le Havre railway in NW France. The contents of the paper is innovative, it is written in a well-structured and mostly clear manner and the research and the results are of high relevance for NHESS. The results of the paper are convincing! I suggest the publication with a few minor amendments and some additional explanations:
We thank Reviewer#1 for his/her positive appraisal of the paper.

2/ Page 2, line 1-5: I think that the options of using a physically based model are written in a rather pessimistic manner. I think that such a model could be applied with a similar data base and with similar results. However, the big difference would be the required time for setting-up such a model and the required computational time. But it could be run with a rather high temporal resolution instead. Maybe you could elaborate a bit more on those differences.
We agree with Reviewer#1 that the comparison of the IRIP model with other types of models is quite short. We will improve the presentation as suggested by Reviewer#1. We also underline that the positioning of the IRIP model already appears in previous publications (Lagadec et al., 2016, 2018). As the focus of the present paper is more on the evaluation methodology of such a model, we had chosen not to discuss too much the interest of the IRIP model, but we agree that it is necessary to elaborate a bit more for a reader not familiar with the IRIP model. One of the major difference between the IRIP model and hydrological models is that the model is a static model and there is no quantitative simulation of discharge (see also answer to comment 3/).

3/ Page 3, introducing the IRIP model: Can you talk a bit on the temporal resolution. I understood it is a quasi-static model, i.e. no temporal resolution. This should be mentioned.
Reviewer#1 is right. There is no temporal resolution in the IRIP model as it is a static model. It will be explicitly stated in the revised version of the manuscript.

Furthermore, please explain if (and possibly how) variability of rainfall in space is considered and how one may approach/ guess the critical rainfall intensity thresholds.
This point is partially discussed in the Discussion section 4.3 but can be further elaborated in the discussion section (see also answer to point 9/). In the methodology section, the use of a rainfall intensity threshold to define the evaluation area is presented as an alternative example to the definition of the study area from a linear transportation network.

I also think that one should mention the question of an appropriate / meaningful spatial resolution here. You discuss this well in the discussion chapter, but you may refer here already to this discussion, because it is essential for model application.
In the revised version of the manuscript, some elements about the appropriate spatial resolution of input data will be added. Note that the IRIP model can be applied with data (in particular DTM) at various resolutions, depending on the objectives of the study. Coarse resolutions can be used when the objective is to get a broad view of sensitive-non sensitive areas over a territory. If higher resolution data are included (eg Lidar DTM data), it is possible to get more precise information and to have explicit representation of linear features such as ditches or roads. In this case, it is not necessary

to provide exogenous information about road networks that are not seen by coarse resolution DTM but are detected with high resolution ones. In this case, adaptation of the model may be required.

4/ Page 3, line39: Typo ("reduced" instead "reduces")
Taken into account

5/ Page 4, line 13: how to guess the "rainfall large than a specified intensity"
See discussion for more details. See also answer to comment 3/.

6/ Page 4, step 2.2.2: Can you explain why the runoff susceptibility map is not important here
We do not say that the runoff susceptibility map to runoff generation is not important, but that it is not used for characterizing the hazard level, when data of localized impacts are used for the model evaluation. Indeed, those data are not directly related to runoff generation process but generally to transfer (erosion and water arrival) and accumulation processes (water stagnation and flooding). This highlights that there are generally no direct observation of runoff and in particular of the generation process. So the map cannot be used for the comparison with localized impact data as the latter do not characterize the generation process. It does not mean that the map has no interest, on the contrary: by pointing out where are the runoff generation areas, the map can suggest mitigation measures to retain runoff where it is produced in order to limit its further transfer and accumulation (see discussion in section 4.4).

7/ Page 5, line 36: I cannot see easily the thalweg structure in figure 4, can you improve this visibility?
The figure will be improved in the revised version.

8/ Page 6, line 31: Figure 5 needs more explanations.
Reviewer#1 is right: the figure caption will be enhanced to better explain the various parts of the figure. On p.6, the reference to the figure was only to illustrate the various types of railway profiles that can be encountered and that are illustrated in the second column of the figure).

9/ Page 11, line 38: Can you elaborate a bit more, if (and if yes, how) this model could account for individual rain storm events. This would be rather interesting.
Since the submission of the paper, further work has been done on analyzing past events and some elements of methodology can be added to the discussion. The approach that has been proposed – in an operational use of the model, not in the perspective of assessing the relevance of the evaluation method like in this paper.
A rain threshold triggering localized impacts can be assessed if a time series of spatialized rain and geo-referenced localized impacts are available for the same storm event. The principle consists in searching for the maximum rainfall for the different impacts, over different durations from 5 minutes to 1 hour. The durations of interest are based on the assumption that the higher the average hazard level that is located in the vicinity of the impact, the lower the amount of rain required to trigger the impact. Initial results points to a relevant duration of 15 minutes to 1 hour. Once the duration has been selected, the minimum rainfall intensity over this duration is selected and considered as the rainfall threshold necessary to trigger all observed incidents. This assumption allows restricting the model's evaluation area to the areas where it has rained enough.

10/ Discussion section: Did you gain any information about possible blockage of culverts / drainage pipes under the railroad track during heavy rainstorms? If yes, it would be very interesting to read about this. I have been informed about such incidences in Germany during after flash floods. Those

created a big problem, when the street or railroad dams were impounded, overflown, eroded and partly broken. I think tis risk is under-estimated if not neglected at all.

Reviewer#1 is right: blockage of culverts/drainage pipes is a common problem in the railway context. In addition to blockage related to a particular intense event, progressive filling of the infrastructure by diffuse sediment transport is also a phenomenon that is often observed and that should require a particular warning. Unfortunately, this information is rarely documented in the reports about the impacts, found in the archives. Thus, it was not possible to consider this information in the evaluation methodology. On the other hand, IRIP map of susceptibility to transfer, by highlighting areas prone to sediment transport can allow management and warning to be concentrated on these areas.

11/ Discussion (or conclusions): can you add a paragraph summing the limitations of the model?

Some elements will be added in the discussion about this point. We can mention limitations related to the methodology itself such as the fact that the model does not provide quantitative estimates of runoff. The other limitation is that the produced susceptibility maps are relative to the study area, as the thresholds that divide the factors maps into areas sensitive or not sensitive to runoff are computed on the study area. Therefore, it is not possible to compare maps from two areas, and if the study area changes a little, the map will also change. There are also limitations related to the application of the model. One of them is the long computing time that are required when large study areas are considered or if the DTM resolution is high. Finally, there are limitations in the evaluation itself as the maps of susceptibility to runoff generation were poorly evaluated, due to the lack of appropriate data.

12/ Table 3: Where can one get these estimates for model parameters from? Are these the default estimates? Can you give some reasonable parameter ranges?

The parameters presented in Table 3 are either obtained using the automatic classification, either from expertise of the IRIP model application in various contexts.

IRIP can be applied everywhere without prior knowledge of runoff over the area, however, the relevance of the maps improves significantly when some parameters are adjusted after a study of the area. Some example of such adjustments are provided below:

- the drained area depending on the level of detail expected at the head of the basin, but it remains between 0.5 and 5 ha. Above 5 ha too much information is lost. In addition, localized impacts of runoff were recorded for catchments of about 1 ha.....

- the break of slope: the threshold depends on the calculation method used to compute the break of slope factor, as the number of pixels over which the factor is computed can be chosen by the user. The number is always an odd number and 3 pixels is the minimum number that must be used. If the number of pixels increases, information about microtopography becomes less accurate. The number of pixels much be adapted according to the resolution of the DTM: for instance 3 to 7 pixels are recommended for a DTM of 25m resolution, and between 9 and 25 for a DTM of 5m resolution.

- The types of land use that are considered favorable to runoff can also be modified. The modification of these parameters and the evaluation of their relevance depends on the expert conducting the study, his knowledge of the area, but also on his objective (precise study, or large mesh for larger territories).

13/ Figure 2: somehow difficult to understand

The figure caption will be expanded to better explain what the figure represents.

14/ Figure 4: Please improve / extend this figure a bit:
- Include an inlet, where you show the location of this region within France
- Names of the river are hardly readable.

- Dry thalwegs difficult to guess.
- Rouen and Le Havre urban areas could be shown?

This figure will be improved following the Reviewer's recommendations.

References

Lagadec, L.-R., Moulin, L., Braud, I., Chazelle, B., and Breil, P.: A surface runoff mapping method for optimizing risk assessment on railways, Safety Science, 110, 253-267, 2018.

Lagadec, L.-R., Patrice, P., Braud, I., Chazelle, B., Moulin, L., Dehotin, J., Hauchard, E., and Breil, P.: Description and evaluation of a surface runoff susceptibility mapping method, Journal of Hydrology, 541, Part A, 495-509, 2016.

---

## Referee Comment (RC2) · Anonymous Referee #2 · 14 Nov 2019

Review of "Evaluation of a model for mapping intense pluvial runoff hazard using proxy data of runoff-related impacts. Application to the railway context." by Braud et al.

The paper focuses on the use and evaluation of the Indicator of Intense Pluvial Runoff (IRIP) method. The method is used to provide susceptibility maps for a railway line in Northern France. The paper is well written and organized. The topic is also of high interest. However, one major issue in the methodology may undermine the results of the entire work. Moreover, the authors should be very careful in not using the same figures and same wording used in their previous works.

**Major comments:**

1) This is the critical issue: In Step 4 is written that "If an area classified at risk has a specific supervision measure or mitigation structures have been built, it is moved from 'false alarm' to 'hit' as the implementation of mitigation measures means that the area was indeed at risk, but that no impact was recorded due to the efficiency of the mitigation measure." Step 4 brings a significant a bias in the evaluation, which can be seen as a unidirectional attempt to improve model performance. If authors would like to use an approach taking into account mitigation measures, they should also do the opposite: "If an area not classified at risk has a measure or structure, and no impact was recorded, it should be moved from 'Correct negative' to 'miss'", or, at least, according to section 2.4, if the area was in the "bad weather tour".
Since this may worsen performance, maybe the authors should either create a different model for areas with mitigation measures, or entirely remove them from the evaluation. The performance boost that results in the last column of Table 5 is due to the inappropriate method mentioned above.
This aspect is crucial for the entire paper, because the "Results" section concludes with this sentence:
"The results in the last column present very encouraging values, highlighting the added value of the IRIP maps, and of the vulnerability and mitigation measures characterization, for the evaluation of the IRIP model."
But the last column is affected by this issue, and this casts a shadow on the entire paper.
2) Figure 1 is the same as Figure 4 already published in Lagadec et al., 2018. I understand that you are using the same method. But if a figure has been already published, this should be mentioned in the paper. Moreover, the description of the IRIP method on page 3 uses exactly the same words used in Lagadec et al., 2018.
Reference:
Lagadec, L.-R., Moulin, L., Braud, I., Chazelle, B., and Breil, P.: A surface runoff mapping method for optimizing risk assessment on railways, Safety Science, 110, 253-267, https://doi.org/10.1016/j.ssci.2018.05.014, 2018.
3) Page 3 lines 36-40: Please specify the classification method used. These lines are very unspecific and the conclusions from these lines are not supported.
4) Page 5 lines 15-16: this statement is incorrect. The results of the chi-square do not demonstrate that the relationship is highly significant, but that it possible to reject the

null hypothesis of independence, because it is unlikely that the null hypothesis of independence is true.

5) Page 10 lines 2-8: the demonstration (or assumption?) that each section had the chance to experience a rare event is obscure to me.

**Minor comments:**

1) Page 3 line 39: reduces->reduced
2) Page 6 line 15: either…or
3) Table 2: please specify also in the table the number of d.o.f for the chi-square test.

---

## Author Comment (AC2) · 20 Dec 2019

**Answers to Reviewer#2 comments**

We thank Reveiwer#2 for his/her comments. The reviewer comments appear in black and the answers appear in blue.

1/ The paper focuses on the use and evaluation of the Indicator of Intense Pluvial Runoff (IRIP) method. The method is used to provide susceptibility maps for a railway line in Northern France. The paper is well written and organized. The topic is also of high interest. However, one major issue in the methodology may undermine the results of the entire work.

We thank Reviewer#2 for his.her encouraging comments regarding the interest of our work. We also thank him.her for his.her suggestions regarding the evaluation methodology. We provide a detailed answer below and how we propose to take the suggestions into account in the revised version of the manuscript.

2/ Moreover, the authors should be very careful in not using the same figures and same wording used in their previous works.

See answer to comment 8/ below.

**3/ Major comments:**

This is the critical issue: In Step 4 it is written that "If an area classified at risk has a specific supervision measure or mitigation structures have been built, it is moved from 'false alarm' to 'hit' as the implementation of mitigation measures means that the area was indeed at risk, but that no impact as recorded due to the efficiency of the mitigation measure." Step 4 brings a significant a bias in the evaluation, which can be seen as a unidirectional attempt to improve model performance. If authors could like to use an approach taking into account mitigation measures, they should also do the opposite: "If an area not classified at risk has a measure or structure, and no impact was recorded, it should be moved from 'Correct negative' to 'miss'", or, at least, according to section 2.4, if the area was in the "bad weather tour".

We thank Reviewer#2 for raising this point. We acknowledge that this point is worth being considered in step 4 of the evaluation methodology, provided that the proxy data can really be related to runoff-related risks. If this is the case, we agree with Reviewer#2 that step 4 of the methodology must be modified and that "If an area not classified at risk has a measure or structure, and no impact was recorded, it should be moved from "Correct negative" to "miss'".

Concerning the mitigations measures considered in our case study, the following comments can be made: 1/ Contrarily to runoff-related impacts, hydraulic works and "bad weather tours" only indicate a potential risk, not a proven risk (i.e. a risk that already happened); 2/ The use of mitigations measures as proxy data must be done with caution. Indeed, the construction of hydraulic works or the design of tours take into account not only the hazard parameter, but also the vulnerability and the criticality of the stake. A "bad weather tour" is preferably designed on a section that is critical regarding the train traffic management, or on sections with known structural weaknesses. "Bad weather tours" are not precise, they often involve long linear of the railway, and the whole sections of the tours may not be relevant to runoff risk. This is why we only used them as explanatory factors of false alarms, and not as elements proving the existence of a risk (implying to move "correct negative" to "miss" if an area was not classified at risk but had a hydraulic work or was part of the "bad weather tour").

Nevertheless, we propose to modify the description of step 4 as follows and to add the above discussion to the discussion section.

"This fourth step is necessary to properly take into account the fact that, if an area is at risk, the stakeholder may have taken mitigation measures that may explain the absence of observed impact. Such mitigation measures are therefore likely to explain a certain amount of false alarms. For instance, mitigation structures can be protection to buildings, retention basins, hydraulic works crossing below transport infrastructures, etc... They can also be resilience actions like a reinforced supervision in case of high rainfall amount warning. If an area classified at risk has a specific supervision measure or mitigation structures have been built, it is moved from 'false alarm' to 'hit' as the implementation of mitigation measures means that the area was indeed at risk, but that no impact was recorded due to the efficiency of the mitigation measure. This step will be referred to Step 4.1 in the following. If mitigation measures can be considered as a reliable source of information regarding runoff risk, the section must also be moved from "correct negative" to "miss" if a mitigation measure is present and the area was not classified at risk (this step will be referred as Step 4.2 in the following. The performance measures are then recomputed, based on the modified contingency tables of Step 4.1 or Step 4.2 is the latter is relevant.

After these four steps, the final values of the quantitative performance measures are obtained, characterizing the performance of the IRIP mapping model. "

Furthermore, in order to investigate the impact of considering Reviewer#2' suggestion in the evaluation, we performed additional computations of the evaluation criteria, taking into account Reviewer#2 suggestion, to see how it modifies the computed performance criteria of the IRIP model. The reference results are those of column 2 in Table 5, where only the vulnerability of the railway is taken into account in the evaluation. We computed the criteria by taking into account mitigation measures as follows: "Stet 4.1": moving from 'false alarm' to 'hit' the sections where a mitigation measure is present but no impact  was recorded (present version of the paper), and  "Step 4.2": moving from 'false alarm' to 'hit' the sections where a mitigation measure is present but no impact was recorded AND moving from "Correct negative" to "miss'" the sections where a mitigation measure is present and the section was not tagged at risk by the IRIP model (Reviewer#2 suggestion). Mitigation measures were accounted for as follows and the results are reported in Table 1 below:
- By taking into account hydraulic works only
- By taking into account "bad weather tours" only
- By taking into account both hydraulic works and "bad weather tours".

When comparing the results obtained in Step 4.1 and Step 4.2, the results of Table 1 show that, in Step 4.2, when only hydraulic works are taken into account, only three additional sections are moved from "correct negative" to "miss", leading to a small decrease of POD. When only "bad weather tours" are considered as mitigation measures in Step 4.2, 9 additional sections are moved from "correct negative" to "miss", leading to a decrease of POD from 94% to 84%. Finally, when hydraulic works and "bad weather tours" are both taken into account, POD decreases from 96% to 86%. FAR remains unchanged when moving from Step 4.1 to Step 4.2.

When comparing results of Step 4.2 with the reference results (without accounting for mitigation measures), Table 1 shows that when considering hydraulic works only, POD slightly decreases whereas FAR is greatly improved. When considering "bad weather tours" only, POD decreases significantly (from 93% to 84%) as the number of "miss" increases. Nevertheless, FAR is improved from 58% to 49%. Finally, when both hydraulic works and "bad weather tours" are accounted for, POD decreases from 96 to 86% but FAR is improved from 58 to 28%.

Even when modifying the evaluation methodology according to Step 4.2, the performance of the IRIP models remains satisfactory (see also answer to comment 6/).

Table 1: Performance criteria assessing the predictive power of the IRIP model in identifying sections with a proven risk using different methods to take into account mitigation measures. In columns 3 to 5, the figures correspond to Step 4.2 (in parenthesis to Step 4.1, i.e. the first version of this paper).

| | Reference IRIP when taking vulnerability into account but not mitigation measures | IRIP when taking vulnerability and hydraulic works into account | IRIP when taking vulnerability and "bad weather tours" into account | IRIP when taking vulnerability hydraulic works and "bad weather tours" into account |
|---|---|---|---|---|
| Number of 'Hit' | 55 | 85 (85) | 67 (67) | 95 (95) |
| Number of 'False Alarm' | 77 | 47 (47) | 65 (65) | 37 (37) |
| Number of 'Correct Negative' | 46 | 43 (46) | 37 (46) | 35 (46) |
| Number of 'Miss' | 4 | 7 (4) | 13 (4) | 15 (4) |
| Probability of Detection: POD (%) | 93 | 92 (96) | 84 (94) | 86 (96) |
| False Alarm Ratio: FAR (%) | 58 | 36 (36) | 49 (49) | 28 (28) |
| $\chi^2$ | 19 | 36.8 (46.1) | 9 (27.9) | 26.7 (59.8) |

Table 2: Performance criteria assessing the predictive power of the presence of hydraulic works or "bad weather tours" in identifying sections with a proven risk using different methods to take into account mitigation measures

|  | Impacts explained by hydraulic works | Impacts explained by "bad weather tours" | Impacts explained by hydraulic works or "bad weather tours" |
|---|---|---|---|
| Number of 'Hit' | 22 | 29 | 40 |
| Number of 'False Alarm' | 33 | 21 | 51 |
| Number of 'Correct Negative' | 90 | 102 | 72 |
| Number of 'Miss' | 37 | 30 | 19 |
| Probability of Detection: POD (%) | 37 | 49 | 68 |
| False Alarm Ratio: FAR (%) | 60 | 42 | 56 |
| $\chi^2$ | 2 (not significant) | 20.6 | 11.1 |

Furthermore, we also computed the performance criteria to assess the predictive power of, respectively, hydraulic works only (Table 2, column 2); "bad weather tours" only (Table 2, column 3); and a combination of both sources of information (hydraulic works OR "bad weather tours") (Table 2, column 4) in explaining the recorded runoff-related impacts. The contingency tables presented in Table 2 were computed assuming that a section was at risk if a hydraulic work or a "bad weather tour" was present in this section (i.e. assuming that they could be considered as proven risk). The results show that the hypothesis that hydraulic works and runoff-related risks are independent cannot be rejected ($\chi^2$ not significant), i.e. hydraulic works have low predictive power about the occurrence of a risk. On the other hand, "bad weather tours" have a predictive power but lower values of POD (49%) than the IRIP model without mitigation measures (93%) (comparison of column 3 of Table 2 and column 2 of Table 1). The FAR value is lower than for the IRIP model without mitigation measures (42% as compared to 58%) but is not so different. Finally, when the presence of hydraulic works and "bad weather tours" are combined (Table 2, column 4), the POD increases to 68% as compared to considering hydraulic works only or "bad weather tours" only. The FAR is 56%, an intermediate value between the one of hydraulic work only (60%) and "bad weather tour" only (42%). In any case, the predictive power of the IRIP model is higher than when considering hydraulic works or "bad weather tours" as proxy for the risk of intense runoff. The results presented in Table 2 also highlight that the "bad weather tour" is a more reliable proxy data for runoff related risk than hydraulic works.

We propose to add these elements to the discussion section.

5/ Since this may worsen performance, maybe the authors should either create a different model for areas with mitigation measures, or entirely remove them from the evaluation.
In our case study, this solution would have the drawback of excluding a large part of the runoff-related impacts from the analysis (22 impacts out of 59 also have a hydraulic works; 29 impacts out of 59 also have a "bad weather tour"), and this would decrease the strength of the analysis.
Instead, we prefer to modify the evaluation methodology as presented in answer to comment 3/ and modify the results accordingly.

6/ The performance boost that results in the last column of Table 5 is due to the inappropriate method mentioned above.

As shown by the additional results provided in the answer to comment 3/, taking into account the revision of the methodology proposed by Reviewer#2 leads to results that are similar to those of the present version of the paper for FAR and are slightly lower for POD. Nevertheless, the performance criteria remain satisfactory and confirm the benefit of the IRIP model in identifying sections at risk. Furthermore, considering that mitigation measures only indicate a potential risk and not a proven risk, the final POD obtained by taking Reviewer#2 suggestion into account is the lowest value that can be expected, as it is the most pessimistic way to take into account information about mitigation measures.

In the revised version of the manuscript, we prefer to modify the evaluation methodology as presented in answer to comment 3/ and modify the results accordingly. In particular, we propose to add columns with the intermediate results presented in answer to comment 3/ so that the reader can appreciate the impact of the way mitigation measures are taken into account in the evaluation methodology. The content of answers to comment 3/ will be presented partly in the Results section and partly in the Discussion section.

7/ This aspect is crucial for the entire paper, because the "Results" section concludes with this sentence:

"The results in the last column present very encouraging values, highlighting the added value of the IRIP maps, and of the vulnerability and mitigation measures characterization, for the evaluation of the IRIP model."

But the last column is affected by this issue, and this casts a shadow on the entire paper.

As shown in the complementary results presented in the answer to comment 3/, a modification of the methodology following Reviewer#2 suggestion does not change dramatically the conclusions of the study and the sentence underlined by Reviewer#2 remains correct, as well as our conclusions that remain supported by the analysis.

Furthermore, we would like to highlight that the present work showed how it was crucial to take into account the information about vulnerability and mitigation measures in the evaluation methodology. In general, only impacts data are considered, because the vulnerability and mitigation measures are more complex to incorporate. In this paper the novelty of the approach is to have included vulnerability and mitigation measures in the methodology. Due to the quality of the data set, we were able to quantify the impact of taking into account or not this information on performance criteria. We propose to add this sentence in the conclusion.

8/ Figure 1 is the same as Figure 4 already published in Lagadec et al., 2018. I understand that you are using the same method. But if a figure has been already published, this should be mentioned in the paper. Moreover, the description of the IRIP method on page 3 uses exactly the same words used in Lagadec et al., 2018. [Lagadec, L.-R., Moulin, L., Braud, I., Chazelle, B., and Breil, P.: A surface runoff mapping method for optimizing risk assessment on railways, Safety Science, 110, 253-267, https://doi.org/10.1016/j.ssci.2018.05.014, 2018. ]

Figure 1 is not exactly the same as the one published in Lagadec et al. (2018). As we believe this figure explains clearly the methodology, we prefer to keep it and to mention that the figure is "adapted from Lagadec et al. (2018)" in the figure caption.

In terms of description of the IRIP method, we have already mentioned in the current version of the paper (p.3 line 15-16) that the provided description is mainly borrowed from Lagadec et al. (2018): "The present description is mainly taken from Lagadec et al. (2018) that retained improvements proposed by Lagadec (2017) to the IRIP model."

We propose to modify the description as follows to make it less similar to that of Lagadec et al. (2018), and to include also the answer to Reviewer#2 comment 9/ and Reviewer#1 comment 3.3/ in the description.

[revised manuscript text omitted]

are not seen by coarse resolution DTM but are detected with high resolution ones. In this case, adaptation of the model may be required."

9/ Page 3 lines 36-40: Please specify the classification method used. These lines are very unspecific and the conclusions from these lines are not supported.
We propose to modify the text as follows:
"To determine the thresholds separating the topographic indicator values (slope and topographic index respectively) into values favorable or not to runoff, an automatic classification, the "K-mean clustering method for grids" provided in SAGA GIS was used. The third option that combines two methods: the iterative minimum distance (Forgy, 1965) and the hill-climbing method (Rubin, 1967) to divide the grid values into two classes was retained. The principle of the method is to maximize the inter-class variance, while minimizing the intra-class variance. As the classification is performed using all the grid points located in the study area, the threshold value, separating the two classes (favorable or not to runoff), depends on the study area. The IRIP model can therefore be applied to various territories without a priori local knowledge on the area, as the thresholds can be automatically computed. If local knowledge about threshold values is available, these threshold values can be specified by the user. Note that the choice of the two thresholds for the slope and topographic index has an impact on four indicators out of the 15 presented in Figure 1."

10/ Page 5 lines 15-16: this statement is incorrect. The results of the chi-square do not demonstrate that the relationship is highly significant, but that it possible to reject the null hypothesis of independence, because it is unlikely that the null hypothesis of independence is true.
The sentence will be modified as follows:
"A value of $\chi^2$ larger than 10.83 shows that the null hypothesis (independence between the risk levels and the IRIP map) can be rejected at the 0.1% level."

11/ Page 10 lines 2-8: the demonstration (or assumption?) that each section had the chance to experience a rare event is obscure to me.
The point mentioned by Reviewer #2 was raised in the manuscript to support the fact that the chosen case study was adequate to assess the relevance of the proposed evaluation methodology. In particular, the evaluation of the methodology would be biased if the duration of data collection was not long enough so that each section of the railway has had the opportunity to be affected by a heavy rainfall event. In this case, the IRIP model could indicate a risk in a section without reported runoff-related impact because no intense rainfall event would have occurred at that location. To show that the hypothesis that each railway section has had an equal opportunity to experience a high rainfall event, we computed the probability of experiencing [resp. not experiencing] rainfall events of several return periods over a duration of 100 years. This probability is (1-(1-0.1)^100) = 0.99997 [resp. less than 0.0001%] for a 10-year return period, (1-(1-0.05)^100) = 0.994 [resp. less than 1%] for a 20-year return period, and (1-(1-0.0.02)^100) = 0.867 [resp. 13%] for a 50-year return period. Therefore, the working hypothesis is valid and we can conclude that our application of the evaluation methodology is not biased and that the case study was adequate to assess the relevance of the proposed evaluation methodology. We propose to reformulate the sentences as follows:

"Another question is: has each section of the railway had the opportunity to be affected by runoff, i.e has each section of the railway had the opportunity to be affected by an intense rainfall event? If it was not the case, the IRIP model could indicate a risk in a section that would not have been impacted in the absence of any intense rainfall event at that location. To assess the validity of this working hypothesis: "each section had the opportunity to be affected by an intense runoff event", we can

calculate the probability of not having experienced a rainfall event of a given return period during one century. This probability is less than 0.001% for a 10-year return period $[(1/10)^{100}]$, less than 1% $[(1/20)^{100}]$ for a 20-year return period, and 13% $[(1/50)^{100}]$ for a 50-year return period respectively. Therefore, it can be assumed that each section of the railway had the opportunity to experience a rare event at least once during the data collection period. This shows that, if the database is long enough and of course comprehensive (i.e. all the occurred runoff-related impacts were properly reported), the working hypothesis can be accepted and therefore, performance measures can be considered as not biased. In the present case, the comprehensiveness of the database is exceptional, but far from being perfect. However, it was the best that could be collected, and the duration of data collection (more than one century) ensures that the chosen case study was relevant to assess the relevance of the proposed evaluation methodology."

**Minor comments:**
12/ Page 3 line 39: reduces->reduced  Will be corrected.
13/ Page 6 line 15: either…or  Will be corrected.
14/ Table 2: please specify also in the table the number of d.o.f for the chi-square test.
The number of degrees of freedom is 1. This will be added to the caption of Table 2.

---

## Author Comment (AC3) · 20 Dec 2019

**Answers to Reviewer#1 comments**

We thank Reveiwer#1 for his/her comments. The reviewer comments appear in black and the answers appear in blue.

1/ This manuscript presents an innovative model regarding the risk of pluvial local flooding and the possible impacts on a traffic infrastructure, in this case a railroad line. The manuscript mainly presents the model concept and an application to the Rouen – Le Havre railway in NW France. The contents of the paper is innovative, it is written in a well-structured and mostly clear manner and the research and the results are of high relevance for NHESS. The results of the paper are convincing! I suggest the publication with a few minor amendments and some additional explanations:

We thank Reviewer#1 for his/her positive appraisal of the paper.

2/ Page 2, line 1-5: I think that the options of using a physically based model are written in a rather pessimistic manner. I think that such a model could be applied with a similar data base and with similar results. However, the big difference would be the required time for setting-up such a model and the required computational time. But it could be run with a rather high temporal resolution instead. Maybe you could elaborate a bit more on those differences.

We agree with Reviewer#1 that the comparison of the IRIP model with other types of models is quite short. We will improve the presentation as suggested by Reviewer#1. We also underline that the positioning of the IRIP model already appears in previous publications (Lagadec et al., 2016, 2018). As the focus of the present paper is more on the evaluation methodology of such a model, we had chosen not to discuss too much the interest of the IRIP model, but we agree that it is necessary to elaborate a bit more for a reader not familiar with the IRIP model. One of the major difference between the IRIP model and hydrological models is that the model is a static model and there is no quantitative simulation of discharge (see also answer to comment 3/).

We propose to modify the sentences as follows:
"As runoff can occur everywhere on a territory, there is a need to provide maps of susceptibility to surface runoff at the scale of a whole territory or an entire transport network. Physically based distributed models may be deployed (e.g. Dabney et al., 2011; Le Bissonnais et al., 2002; Schmocker-Fackel et al., 2007, Smith et al., 1995) have the ability to provide the spatial and temporal evolution of runoff dynamics (water depth and sometimes velocity). However, they require many input data for their set up and calibration that may not be available everywhere. Thus this kind of model may be difficult to deploy on large territories. An alternative solution, called IRIP….."

3.1/ Page 3, introducing the IRIP model: Can you talk a bit on the temporal resolution. I understood it is a quasi-static model, i.e. no temporal resolution. This should be mentioned.

Reviewer#1 is right. There is no temporal resolution in the IRIP model as it is a static model. It will be explicitly stated in the revised version of the manuscript.

"An alternative solution, called IRIP for "Indicator of Intense Pluvial Runoff" was proposed by Dehotin and Breil (2011) for mapping the susceptibility to surface runoff. The model is a score method that provides 3 maps of runoff susceptibility to runoff generation, transfer and accumulation, with 6 susceptibility levels from 0 to 5. The maps are static, therefore, the IRIP model does not have any temporal resolution and the maps do not provide quantitative information about runoff dynamics. The IRIP model allows the creation of three maps representing three different phases of the surface runoff phenomenon: generation, transfer, and accumulation."

3.2/ Furthermore, please explain if (and possibly how) variability of rainfall in space is considered and how one may approach/ guess the critical rainfall intensity thresholds.
This point is partially discussed in the Discussion section 4.3 but can be further elaborated in the discussion section (see also answer to point 9/). In the methodology section, the use of a rainfall intensity threshold to define the evaluation area is presented as an alternative example to the definition of the study area from a linear transportation network.

3.3/ I also think that one should mention the question of an appropriate / meaningful spatial resolution here. You discuss this well in the discussion chapter, but you may refer here already to this discussion, because it is essential for model application.
In the revised version of the manuscript, some elements about the appropriate spatial resolution of input data will be added. Note that the IRIP model can be applied with data (in particular DTM) at various resolutions, depending on the objectives of the study. Coarse resolutions can be used when the objective is to get a broad view of sensitive-non sensitive areas over a territory. If higher resolution data are included (eg Lidar DTM data), it is possible to get more precise information and to have explicit representation of linear features such as ditches or roads. In this case, it is not necessary to provide exogenous information about road networks that are not seen by coarse resolution DTM but are detected with high resolution ones. In this case, adaptation of the model may be required. We propose to modify the end of section 2.1 as follows, taking also into account comment 9/ of Reviewer#1:
"The resolution of the susceptibility maps retains the resolution of the Digital Elevation Model (rasterized topography map) used as input data. To determine the thresholds separating the topographic indicator values (slope and topographic index respectively) into values favorable or not to runoff, an automatic classification, the "K-mean clustering method for grids" provided in SAGA GIS was used. The third option that combines two methods: the iterative minimum distance (Forgy, 1965) and the hill-climbing method (Rubin, 1967) to divide the grid values into two classes was retained. The principle of the method is to maximize the inter-class variance, while minimizing the intra-class variance. As the classification is performed using all the grid points located in the study area, the threshold value, separating the two classes (favorable or not to runoff), depends on the study area. The IRIP model can therefore be applied to various territories without a priori local knowledge on the area, as the thresholds can be automatically computed. If local knowledge about threshold values is available, these threshold values can be specified by the user. Note that the choice of the two thresholds for the slope and topographic index has an impact on four indicators out of the 15 presented in Figure 1. Note also that If higher resolution data are included (e.g. Lidar DTM data), it is possible to get information that is more precise and to have explicit representation of linear features such as ditches or roads. In this case, it is not necessary to provide exogenous information about road networks that are not seen by coarse resolution DTM but are detected with high resolution ones. In this case, adaptation of the model may be required."

4/ Page 3, line39: Typo ("reduced" instead "reduces")
Taken into account

5/ Page 4, line 13: how to guess the "rainfall large than a specified intensity"
See discussion for more details. See also answer to comment 3/.

6/ Page 4, step 2.2.2: Can you explain why the runoff susceptibility map is not important here
We do not say that the runoff susceptibility map to runoff generation is not important, but that it is not used for characterizing the hazard level, when data of localized impacts are used for the model

evaluation. Indeed, those data are not directly related to runoff generation process but generally to transfer (erosion and water arrival) and accumulation processes (water stagnation and flooding). This highlights that there are generally no direct observation of runoff and in particular of the generation process. So the map cannot be used for the comparison with localized impact data as the latter do not characterize the generation process. It does not mean that the map has no interest, on the contrary: by pointing out where are the runoff generation areas, the map can suggest mitigation measures to retain runoff where it is produced in order to limit its further transfer and accumulation (see discussion in section 4.4).

We propose to modify step 2.2.2 as follows:

"As the IRIP model provides three maps, it also means choosing the maps that will be considered in the evaluation. The susceptibility map to runoff generation is not used for characterizing the hazard level, when data of localized impacts are used for the model evaluation. Previous experience (Lagadec et al., 2016b; 2018) showed that, when compared to localized impact data, susceptibility maps to transfer and accumulation were relevant, with the susceptibility map to transfer generally associated with erosion, and the susceptibility map to accumulation associated to sediment deposition and flooding.  Note that the same area can have at the same time a high susceptibility to runoff transfer and/or accumulation and/or generation. On the other hand, impact data are not directly related to runoff generation process. Therefore, a composite of the susceptibility maps to runoff transfer and accumulation was built for defining the hazard. This map is the union of the susceptibility maps to transfer and accumulation, i.e. each pixel retains the maximum level of both maps."

7/ Page 5, line 36: I cannot see easily the thalweg structure in figure 4, can you improve this visibility?
The figure was improved in the revised version (see answer to comment 14/.

8/ Page 6, line 31: Figure 5 needs more explanations.
Reviewer#1 is right: the figure caption will be enhanced to better explain the various parts of the figure. On p.6, the reference to the figure was only to illustrate the various types of railway profiles that can be encountered and that are illustrated in the second column of the figure).
The new caption reads as follows
"Figure 5: Vulnerability tree of the railway based on expert judgment. Each column corresponds to one criteria considered when computing the vulnerability: the exposure (non-exposed sections are long tunnels or viaducts) (column 1)", the type of railway profile (column 2), the length of the section (column 3) and the existence of a singularity (either level crossings, road bridges or tunnel inlets or outlets) (column 4). The +1 in the red circles indicate that 1 is added to the vulnerability score of the section to provide the final score that appears in the last column of the figure."

9/ Page 11, line 38: Can you elaborate a bit more, if (and if yes, how) this model could account for individual rain storm events. This would be rather interesting.
Since the submission of the paper, further work has been done on analyzing past events and some elements of methodology can be added to the discussion. The approach that has been proposed – in an operational use of the model, -not in the perspective of assessing the relevance of the evaluation method like in this paper is the following.
A rain threshold triggering localized impacts can be assessed if a time series of spatialized rain and geo-referenced localized impacts are available for the same storm event. The principle consists in searching for the maximum rainfall for the different impacts, over different durations from 5 minutes to 1 hour. The durations of interest are based on the assumption that the higher the average hazard level that is located in the vicinity of the impact, the lower the amount of rain required to trigger the

impact. Initial results points to a relevant duration of 15 minutes to 1 hour. Once the duration has been selected, the minimum rainfall intensity over this duration is selected and considered as the rainfall threshold necessary to trigger all observed incidents. This assumption allows restricting the model's evaluation area to the areas where it has rained enough.
We propose to add this paragraph to the discussion.

10/ Discussion section: Did you gain any information about possible blockage of culverts / drainage pipes under the railroad track during heavy rainstorms? If yes, it would be very interesting to read about this. I have been informed about such incidences in Germany during after flash floods. Those created a big problem, when the street or railroad dams were impounded, overflown, eroded and partly broken. I think tis risk is under-estimated if not neglected at all.
Reviewer#1 is right: blockage of culverts/drainage pipes is a common problem in the railway context. In addition to blockage related to a particular intense event, progressive filling of the infrastructure by diffuse sediment transport is also a difficulty, since there is a large number of small hydraulic works that are difficult to maintain. Unfortunately, the information is rarely documented in the reports about the impacts, found in the archives. Thus, it was not possible to consider this information in the evaluation methodology. On the other hand, IRIP map of susceptibility to transfer, by highlighting areas prone to sediment transport can allow management and warning to be concentrated on these areas.
We propose to add these elements to the discussion.

11/ Discussion (or conclusions): can you add a paragraph summing the limitations of the model?
Some elements will be added in the discussion about this point. We can mention limitations related to the methodology itself such as the fact that the model does not provide quantitative estimates of runoff. The other limitation is that the produced susceptibility maps are relative to the study area, as the thresholds that divide the factors maps into areas sensitive or not sensitive to runoff are computed on the study area. Therefore, it is not possible to compare maps from two areas, and if the study area changes a little, the map will also change. There are also limitations related to the application of the model. The IRIP map of transfer of accumulation strongly depend on the DEM quality, since 3.2 and 4.2 indicators over 5 are derived from the topography. The required computing time is large when large study areas are considered or if the DTM resolution is high. Finally, there are limitations in the evaluation itself as the maps of susceptibility to runoff generation were poorly evaluated, due to the lack of appropriate data.

12/ Table 3: Where can one get these estimates for model parameters from? Are these the default estimates? Can you give some reasonable parameter ranges?
The parameters presented in Table 3 are either obtained using the automatic classification, either from expertise of the IRIP model application in various contexts.
IRIP can be applied everywhere without prior knowledge of runoff over the area, however, the relevance of the maps improves significantly when some parameters are adjusted after a study of the area. Some example of such adjustments are provided below:
- the drained area depending on the level of detail expected at the head of the basin, but it remains between 0.5 and 5 ha. Above 5 ha too much information is lost. In addition, localized impacts of runoff were recorded for catchments of about 1 ha.
- the break of slope: the threshold depends on the calculation method used to compute the break of slope factor, as the number of pixels over which the factor is computed can be chosen by the user. The number is always an odd number and 3 pixels is the minimum number that must be used. If the number of pixels increases, information about microtopography becomes less accurate. The number

of pixels must be adapted according to the resolution of the DTM: for instance 3 to 7 pixels are recommended for a DTM of 25m resolution, and between 9 and 25 for a DTM of 5m resolution.
- The types of land use that are considered favorable to runoff can also be modified, for example according to the agriculture cycle on a same plot. The modification of these parameters and the evaluation of their relevance depends on the expert conducting the study, his knowledge of the area, but also on his objective (precise study, or large mesh for larger territories).
We propose to add these elements to the description of step 1 in section 2.4.

13/ Figure 2: somehow difficult to understand
The figure caption will be modified as follows:
"Figure 2: Scheme of the evaluation methodology to assess the relevance of susceptibility maps to runoff, using localized runoff-related impacts proxy data. The grey boxes indicate the information that is used in the various steps of the methodology. Yellow circles presents the various steps of the evaluation methodology leading to the final quantitative evaluation (orange box)"
14/ Figure 4: Please improve / extend this figure a bit:
- Include an inlet, where you show the location of this region within France
- Names of the river are hardly readable.
- Dry thalwegs difficult to guess.
- Rouen and Le Havre urban areas could be shown?

This figure was improved following the Reviewer's recommendations and the caption will be modified as follows:
"Figure 4: Map of the study area in Normandy (northern France). The yellow contour is the boundary of the catchments intercepted by the Rouen- Le Havre railway (line in black and white). Blue lines are the permanent river courses. One can note the dense network of dry talwegs (darker on the DEM) upstream de rivers that can be activated during a rainfall event."

[Figure]

References
Dabney, S.M., Yoder, D.C., Vieira, D.A.N., Bingner, R.L., 2011. Enhancing RUSLE to include runoff-driven phenomena. Hydrol. Process. 25, 1373–1390. http://dx.doi.org/10.1002/hyp.7897.
Forgy, E., 1965. Cluster Analysis of multivariate data: efficiency vs. interpretability of classifications, Biometrics, 21, 768-780.

Lagadec, L.-R., Moulin, L., Braud, I., Chazelle, B., and Breil, P.: A surface runoff mapping method for optimizing risk assessment on railways, Safety Science, 110, 253-267, 2018.

Lagadec, L.-R., Patrice, P., Braud, I., Chazelle, B., Moulin, L., Dehotin, J., Hauchard, E., and Breil, P.: Description and evaluation of a surface runoff susceptibility mapping method, Journal of Hydrology, 541, Part A, 495-509, 2016.

Le Bissonnais, Y., Montier, C., Jamagne, M., Daroussin, J., and King, D.: Mapping erosion risk for cultivated soil in France, CATENA, 46, 207-220, https://doi.org/10.1016/S0341-8162(01)00167-9, 2002.

Rubin, J., 1967. Optimal Classification into Groups: An Approach for Solving the Taxonomy Problem, J. Theoretical Biology, 15,103-144.

Schmocker-Fackel, P., Naef, F., and Scherrer, S.: Identifying runoff processes on the plot and catchment scale, Hydrology and Earth System Sciences, 11, 891-906, 2007.

Smith, R.E., Goodrich, D.C., Woolhiser, D.A., Unkrich, C.L., 1995. KINEROS – a kinematic runoff and erosion model. Comput. Models Watershed Hydrol. 20, 627–668.

---

## Author Response (AR1)

We thank both reviewers for their comments. They allowed a deeper analysis of the results obtained in the study and contributed to increase the relevance of the results. The reviewers' comments were taken into account in the revised version of the manuscript, as explained below. The reviewers' comments appear in black and the answers appear in blue. The lines numbers refer to the lines numbers of the revised manuscript.

5

**Answers to Reviewer#1 comments**

1/ This manuscript presents an innovative model regarding the risk of pluvial local flooding and the possible impacts on a traffic infrastructure, in this case a railroad line. The manuscript mainly presents the model concept and an application to the Rouen – Le Havre railway in NW France.

The contents of the paper is innovative, it is written in a well-structured and mostly clear manner and the research and the results are of high relevance for NHESS. The results of the paper are convincing!

I suggest the publication with a few minor amendments and some additional explanations:

We thank Reviewer#1 for his/her positive appraisal of the paper.

**15**

35

10

2/ Page 2, line 1-5: I think that the options of using a physically based model are written in a rather pessimistic manner. I think that such a model could be applied with a similar data base and with similar results. However, the big difference would be the required time for setting-up such a model and the required computational time. But it could be run with a rather high temporal resolution instead. Maybe you could elaborate a bit more on those differences.

20 We agree with Reviewer#1 that the comparison of the IRIP model with other types of models was quite short. We improved the presentation as suggested by Reviewer#1. One of the major difference between the IRIP model and hydrological models is that the model is a static model and there is no quantitative simulation of discharge (see also answer to comment 3/).

We modified the sentences as follows:

- 25 "As runoff can occur everywhere on a territory, there is a need to provide maps of susceptibility to surface runoff at the scale of a whole territory or an entire transport network. Physically based distributed models may be deployed (e.g. Dabney et al., 2011; Le Bissonnais et al., 2002; Schmocker-Fackel et al., 2007, Smith et al., 1995). They have the ability to provide the spatial and temporal evolution of runoff dynamics (water depth and sometimes velocity). However, they require many input data for their set up and calibration that may not be available everywhere. Thus, this kind of model may be difficult to deploy on large
- 30 territories. An alternative solution, ...." [lines 38-40 p.1 and 1-3 p.2]

3.1/ Page 3, introducing the IRIP model: Can you talk a bit on the temporal resolution. I understood it is a quasi-static model, i.e. no temporal resolution. This should be mentioned.

Reviewer#1 is right. There is no temporal resolution in the IRIP model as it is a static model. It is now explicitly stated in the revised version of the manuscript.

"IRIP maps are static, therefore, the IRIP model does not have any temporal resolution and the maps do not provide quantitative information about runoff dynamics. However, the maps remain useful for prevention purpose, provided they are properly evaluated." [lines 8-10 p.2]

40 3.2/ Furthermore, please explain if (and possibly how) variability of rainfall in space is considered and how one may approach/ guess the critical rainfall intensity thresholds. This point is partially discussed in the Discussion section 4.3 but was further elaborated in the discussion section (see also answer to point 9/). In the methodology section, the use of a rainfall intensity threshold to define the evaluation area is only presented as an illustrative example for the definition of the study area from a linear transportation network.

- 5 3.3/ I also think that one should mention the question of an appropriate / meaningful spatial resolution here. You discuss this well in the discussion chapter, but you may refer here already to this discussion, because it is essential for model application. In the revised version of the manuscript, some elements about the appropriate spatial resolution of input data were added. Note that the IRIP model can be applied with data (in particular DTM) at various resolutions, depending on the objectives of the study. Coarse resolutions can be used when the objective is to get a broad view of sensitive-non sensitive areas over a territory.
- 10 If higher resolution data are included (eg Lidar DTM data), it is possible to get more precise information and to have explicit representation of linear features such as ditches or roads that improve the representation of water pathways. The end of section 2.1 was modified as follows, taking also into account comment 9/ of Reviewer#1: "The resolution of the susceptibility maps retains the resolution of the Digital Elevation Model (rasterized topography map) used as input data. To determine the thresholds separating the topographic indicator values (slope and topographic index)
- 15 respectively) into values favorable or not to runoff, an automatic classification, the K-mean clustering method for grids provided in SAGA GIS was used. The third option of the function that combines two methods: the iterative minimum distance (Forgy, 1965) and the hill-climbing method (Rubin, 1967) to divide the grid values into two classes was used. The principle of the method is to maximize the inter-class variance, while minimizing the intra-class variance. As the classification is performed using all the grid points located in the study area, the threshold value, separating the two classes (favorable or not
- 20 to runoff), depends on the study area. The IRIP model can therefore be applied to various territories without a priori local knowledge on the area, as the thresholds can be automatically computed. If local knowledge about threshold values is available, the user can alternatively specify these threshold values." [lines 39-41 p.3 and 1-9 p.4]

4/ Page 3, line39: Typo ("reduced" instead "reduces")

25 Taken into account

5/ Page 4, line 13: how to guess the "rainfall larger than a specified intensity" See section 4.3 of the discussion [lines 28-39 p 14] for more details. See also answer to comment 9/.

30 6/ Page 4, step 2.2.2: Can you explain why the runoff susceptibility map is not important here

We do not say that the runoff susceptibility map to runoff generation is not important, but that it is not used for characterizing the hazard level, when data of localized impacts are used for the model evaluation. Indeed, those data are not directly related to runoff generation process but generally to transfer (erosion and water arrival) and accumulation processes (water stagnation and flooding). This highlights that there are generally no direct observation of runoff and in particular of the generation process.

35 So the map cannot be used for the comparison with localized impact data as the latter do not characterize the generation process. It does not mean that the map has no interest, on the contrary: by pointing out where the runoff generation areas are localized, the map can suggest mitigation measures to retain runoff where it is produced in order to limit its further transfer and accumulation (see discussion in section 4.4).

We modified the description of step 2.2.2 as follows:

40 "As the IRIP model provides three maps, it also means choosing the maps that will be considered in the evaluation. Previous experience (Lagadec et al., 2016b; 2018) showed that, when compared to localized impact data, susceptibility maps to transfer and accumulation were relevant, with the susceptibility map to transfer generally associated with erosion, and the susceptibility map to accumulation associated to sediment deposition and flooding. On the other hand, localized runoff related impact data

are not directly related to the runoff generation process. In such conditions, the susceptibility map to runoff generation cannot be used for characterizing the hazard level, and a composite of the susceptibility maps to runoff transfer and accumulation is used to define the hazard. " [lines 34-38 p.4 and 1-2 p.5]

5 7/ Page 5, line 36: I cannot see easily the thalweg structure in figure 4, can you improve this visibility? The figure was improved in the revised version (see answer to comment 14/.)

8/ Page 6, line 31: Figure 5 needs more explanations.

Reviewer#1 is right: the figure caption was enhanced to better explain the various parts of the figure. On p.6, the reference tothe figure was only to illustrate the various types of railway profiles that can be encountered and that are illustrated in the second column of the figure).

The new caption reads as follows

40

"Figure 5: Vulnerability tree of the railway sections (also called earthworks) based on expert judgment. Each column corresponds to one criteria considered when computing the vulnerability of the section: the exposure (non-exposed sections

- 15 are long tunnels or viaducts) (column 1)", the type of railway profile (column 2), the length of the section (column 3) and the existence of a singularity (either level crossings, road bridges or tunnel inlets or outlets) (column 4). The +1 in the red circles indicate that 1 is added to the vulnerability score of the section to provide the final score that appears in the last column of the figure."
- 20 9/ Page 11, line 38: Can you elaborate a bit more, if (and if yes, how) this model could account for individual rain storm events.This would be rather interesting.

Since the submission of the paper, further work has been done on analyzing past events and some elements of methodology were added to the discussion. The approach that was proposed – in an operational use of the model, -not in the perspective of assessing the relevance of the evaluation method like in this paper is the following.

- 25 "The evaluation area can be defined using rainfall data and a given rainfall intensity threshold. Given that the event are often much localized, the use of radar rainfall data with a short time step (e.g. 5 min time step) is recommended, as shown by Marra et al. (2016) for landslides. The rainfall threshold triggering localized impacts can be assessed if a time series of spatialized rain and geo-referenced localized impacts are available for the same storm event. The principle consists in searching for the maximum rainfall for the different impacts, over different durations from 5 minutes to 1 hour. The durations of interest are
- 30 based on the assumption that the higher the average hazard level that is located in the vicinity of the impact, the lower the amount of rain required to trigger the impact. Initial results points to a relevant duration of 15 minutes to 1 hour. Once the duration has been selected, the minimum rainfall intensity over this duration is selected and considered as the rainfall threshold necessary to trigger all observed impacts. This assumption allows restricting the model's evaluation area to the areas where it has rained enough. If data collection about impacts is comprehensive and information about protection infrastructure is
  26 and EAD and EAD and EAD are the average has a server at with a need a server at 20 and 50 and 50
- available, POD and FAR values can be computed with a good accuracy." [lines 38-42 p.14 and 1-7 p.15]

10/ Discussion section: Did you gain any information about possible blockage of culverts / drainage pipes under the railroad track during heavy rainstorms? If yes, it would be very interesting to read about this. I have been informed about such incidences in Germany during after flash floods. Those created a big problem, when the street or railroad dams were impounded, overflown, eroded and partly broken. I think tis risk is under-estimated if not neglected at all.

Reviewer#1 is right: blockage of culverts/drainage pipes is a common problem in the railway context. In addition to blockage related to a particular intense event, progressive filling of the infrastructure by diffuse sediment transport is also a difficulty, since there is a large number of small hydraulic works that are difficult to maintain. Unfortunately, the information is rarely

documented in the reports about the impacts, found in the archives. Thus, it was not possible to consider this information in the evaluation methodology. On the other hand, IRIP map of susceptibility to transfer, by highlighting areas prone to sediment transport can allow management and warning to be concentrated on these areas. These elements were added to the discussion section 4.1.

- 5 "One of the reason for this low predictive power of hydraulic works could be the following. Blockage of culverts/drainage pipes is a common problem in the railway context. In addition to blockage related to a particular intense event, progressive filling of the infrastructure by diffuse sediment transport is also a difficulty, since there is a large number of small hydraulic works that are difficult to maintain. Unfortunately, the information about blockage of hydraulic works is rarely documented in the reports about the impacts, found in the archives. On the other hand, other hydraulic works are well dimensioned and very
- 10 efficient. Thus, hydraulic works can sometimes increase the vulnerability and sometimes decrease it. Therefore, it was not possible to consider this information as a reliable source of information for proven risk in the evaluation methodology nor in the vulnerability tree. On the other hand, the IRIP map of susceptibility to transfer, by highlighting areas prone to sediment transport can allow management and warning to be concentrated on these areas."[lines 34-42 p11 and 1-2 p12]
- 15 11/ Discussion (or conclusions): can you add a paragraph summing the limitations of the model?

The following elements were added to the discussion section 4.1.

"There are limitations related to the IRIP model itself, the main one being that the model does not provide quantitative estimates of runoff. The other limitation is that the produced susceptibility maps are relative to the study area, as the thresholds that divide the factors maps into areas sensitive or not sensitive to runoff are computed on the study area. Therefore, it is not

- 20 possible to compare maps from two areas, and if the study area changes a little, the map will also change. There are also limitations related to the application of the model. The IRIP map of transfer of accumulation strongly depend on the DEM quality, since 3 and 4 indicators over 5 are derived from the topography. The required computing time is large when large study areas are considered or if the DTM resolution is high. Finally, there are limitations in the evaluation itself as the maps of susceptibility to runoff generation were poorly evaluated, due to the lack of appropriate data."
- 25 [lines 31-38 p.13]

30

12/ Table 3: Where can one get these estimates for model parameters from? Are these the default estimates? Can you give some reasonable parameter ranges?

The parameters presented in Table 3 are either obtained using the automatic classification, either from expertise of the IRIP model application in various contexts.

IRIP can be applied everywhere without prior knowledge of runoff over the area, however, the relevance of the maps improves significantly when some parameters are adjusted after a study of the area.

We added some elements in section 2.3.2 [lines 24-26 p.6]

"The IRIP model parameterization used in this study is presented in Table 3. Given the low local knowledge of the study area, the thresholds defining the classes favorable or not favorable to runoff were computed using the classification method proposed by default in the IRIP model, contrarily to the application by Lagadec et al. (2018) that used values derived from local expertise. Thresholds for topographic index and slope indicators were therefore defined using the classification method. The threshold of the drained area indicator was fixed to 2.5 ha following sensitivity tests performed by Lagadec (2017) to identify, in this specific catchment, a minimum surface from which significant surface runoff can be generated. Other thresholds for soil depth,

40 hydraulic conductivity, slacking, erodibility, upslope area sensitive to runoff generation were chosen as in Lagadec et al. (2018)."

And the impact of the choice of the parameters is further discussed in section 4.2 [lines 15-30 p.13]:

"Results of the evaluation method also depend on the values of the parameters chosen to compute the three susceptibility maps (as specified in Table 1). The IRIP model can be applied everywhere without prior knowledge of runoff over the area. However, the relevance of the maps improves significantly when some parameters are adjusted using local knowledge about the area. Examples of such adjustments can be the following:

- The drained area threshold depends on the level of detail expected at the head of the basin, but it remains between 0.5 and 5 ha. Above 5 ha too much information is lost (Lagadec, 2017). Note also that localized impacts of runoff were recorded for catchments of a few hectares and this consideration also guided the choice of the threshold value.
  - The break of slope threshold depends on the calculation method used to compute the break of slope factor, as the user can chose the number of pixels over which the factor is computed. The number is always an odd number and 3 pixels is the minimum number that must be used. If the number of pixels increases, information about micro-topography becomes less accurate. The number of pixels must be adapted according to the resolution of the DTM: for instance, 3 to 7 pixels are recommended for a DTM of 25 m resolution, and between 9 and 25 for a DTM of 5 m resolution.
  - The types of land use that are considered favorable to runoff can also be modified, for example according to the agriculture cycle on a same plot. The modification of these parameters and the evaluation of their relevance depends on the expert conducting the study, his.her knowledge of the area, but also on his.her objective (precise study, or large mesh for larger territories)."

**13/ Figure 2: somehow difficult to understand**

20 The figure caption was modified as follows:

"Figure 2: Scheme of the evaluation methodology to assess the relevance of susceptibility maps to runoff, using localized runoff-related impacts proxy data. The grey boxes indicate the information that is used in the various steps of the methodology. Yellow circles presents the various steps of the evaluation methodology leading to the final quantitative evaluation (orange box)."

**25**

10

15

14/ Figure 4: Please improve / extend this figure a bit:

- Include an inlet, where you show the location of this region within France
- Names of the river are hardly readable.
- Dry thalwegs difficult to guess.
- Rouen and Le Havre urban areas could be shown?

This figure was improved following the Reviewer's recommendations and the caption was modified as follows:

"Figure 4: Map of the study area in Normandy (northern France). The yellow contour is the boundary of the catchments intercepted by the Rouen- Le Havre railway (line in black and white). Blue lines are the permanent river courses. One can note the dense network of dry talwegs (darker on the DEM) upstream de rivers that can be activated during a rainfall event."

**35**

30

**Answers to Reviewer#2 comments**

1/ The paper focuses on the use and evaluation of the Indicator of Intense Pluvial Runoff (IRIP) method. The method is used to provide susceptibility maps for a railway line in Northern France. The paper is well written and organized. The topic is also of high interest. However, one major issue in the methodology may undermine the results of the entire work.

40

We thank Reviewer#2 for his.her encouraging comments regarding the interest of our work. We also thank him.her for his.her suggestions regarding the evaluation methodology. We provide a detailed answer below and we took the suggestions into account in the revised version of the manuscript and the main remark of the reviewer allowed us to deepen the analysis of our results and to improve the robustness of the conclusions of the paper.

45

2/ Moreover, the authors should be very careful in not using the same figures and same wording used in their previous works. See answer to comment 8/ below.

**3/ Major comments:**

This is the critical issue: In Step 4 it is written that "If an area classified at risk has a specific supervision measure or mitigation structures have been built, it is moved from 'false alarm' to 'hit' as the implementation of mitigation measures means that the

- 5 area was indeed at risk, but that no impact as recorded due to the efficiency of the mitigation measure." Step 4 brings a significant a bias in the evaluation, which can be seen as a unidirectional attempt to improve model performance. If authors could like to use an approach taking into account mitigation measures, they should also do the opposite: "If an area not classified at risk has a measure or structure, and no impact was recorded, it should be moved from 'Correct negative' to 'miss'", or, at least, according to section 2.4, if the area was in the "bad weather tour".
- 10 We thank Reviewer#2 for raising this point. We acknowledge that this point is worth being considered in step 4 of the evaluation methodology, provided that the proxy data can really be related to runoff-related risks. If this is the case, we agree with Reviewer#2 that step 4 of the methodology must be modified and that "If an area not classified at risk has a measure or structure, and no impact was recorded, it should be moved from "Correct negative" to "miss".

We modified the description of step 4 as follows [lines 33-42 p.5 and 1-5 p.6]:

- 15 "This fourth step is necessary to properly take into account the fact that, if an area is at risk, the stakeholder may have taken mitigation measures that may explain the absence of observed impact. Such mitigation measures are therefore likely to explain a certain amount of false alarms. For instance, mitigation structures can be protection to buildings, retention basins, hydraulic works crossing below transport infrastructures, etc... They can also be resilience actions like a reinforced supervision in case of high rainfall amount warning. Their aim is to reduce damage consequences by issuing early warning or by performing local
- 20 work to reduce potential damages during an event. If an area classified at risk has a specific supervision measure or mitigation structures have been built, it is moved from 'false alarm' to 'hit' as the implementation of mitigation measures means that the area was indeed at risk, but that no impact was recorded due to the efficiency of the mitigation measure. This step will be referred to Step 4.1 in the following. If mitigation measures can be considered as a reliable source of information regarding runoff risk, the section must also be moved from 'correct negative' to 'miss' if a mitigation measure is present and the area
- 25 was not classified at risk (this step will be referred as Step 4.2 in the following). The performance measures are then recomputed, based on the modified contingency tables of Step 4.1 or Step 4.2 if the latter is relevant. After these four steps, the final values of the quantitative performance measures are obtained, characterizing the performance of the IRIP mapping model. "
- 30 Furthermore, in order to investigate the impact of considering Reviewer#2' suggestion in the evaluation, we performed additional computations of the evaluation criteria, taking into account Reviewer#2 suggestion, to see how it modifies the computed performance criteria of the IRIP model (see modified step 4 in section 2.4 [lines 21-35 p.8])
  "Step 4: Mitigation measures were considered in a second step. Structural and non-structural mitigation measures were
- considered and inventoried along the whole railway. Structural measures include all the hydraulic structures (drainage structures along or below the railway track, retention ponds, etc...) that were built to help water flow circulation. At SNCF, non-structural measures include surveillance patrols in case of bad weather. These patrols target as a priority the sections registered in what is called the 'bad weather tours'. The latter are defined using local knowledge about the hazard exposure or about the specific infrastructure vulnerabilities. They provide increased and targeted monitoring in case of bad weather and early response if needed.
- 40 We modified the computation of the performance criteria by taking into account mitigation measures (presence of a hydraulic structure in one section, or section registered in the 'bad weather tour') as follows. In Step 4.1, we moved the sections where a mitigation measure is present but no impact was recorded from 'false alarm' to 'hit'. In Step 4.2, we moved the sections where a mitigation measure is present but no impact was recorded from 'false alarm' to 'hit' AND moved the sections where

[revised manuscript text omitted]

4/ Since this may worsen performance, may be the authors should either create a different model for areas with mitigation measures, or entirely remove them from the evaluation.

35

40

In our case study, this solution would have the drawback of excluding a large part of the runoff-related impacts from the analysis (22 impacts out of 59 also have a hydraulic works; 29 impacts out of 59 also have a "bad weather tour"), and this would decrease the strength of the analysis.

Instead, we preferred to modify the evaluation methodology as presented in answer to comment 3/ and to modify the results accordingly.

5/ The performance boost that results in the last column of Table 5 is due to the inappropriate method mentioned above.

As shown by the additional results provided in the answer to comment 3/, taking into account the revision of the methodology proposed by Reviewer#2 leads to results that are similar to those of the present version of the paper for FAR and are slightly lower for POD. Nevertheless, the performance criteria remain satisfactory and confirm the benefit of the IRIP model in identifying sections at risk. Furthermore, considering that mitigation measures only indicate a potential risk and not a proven

risk, the final POD obtained by taking Reviewer#2 suggestion into account is the lowest value that can be expected, as it is the most pessimistic way to take into account information about mitigation measures.

In the revised version of the manuscript, we preferred to modify the evaluation methodology as presented in answer to comment 3/ and we modified the results accordingly. In particular, we modified Table 5 as presented in answer to comment 3/ so that the reader can appreciate the impact of the way mitigation measures are taken into account in the evaluation methodology. See detailed answer to comment 3/

10 detailed answer to comment 3/

6/ This aspect is crucial for the entire paper, because the "Results" section concludes with this sentence:

"The results in the last column present very encouraging values, highlighting the added value of the IRIP maps, and of the vulnerability and mitigation measures characterization, for the evaluation of the IRIP model."

15 But the last column is affected by this issue, and this casts a shadow on the entire paper.

As shown in the complementary results presented in the answer to comment 3/, a modification of the methodology following Reviewer#2 suggestion does not change dramatically the conclusions of the study and the sentence underlined by Reviewer#2 remains correct, as well as our conclusions that remain supported by the analysis.

Furthermore, we would like to highlight that the present work showed how it was crucial to take into account the information about vulnerability and mitigation measures in the evaluation methodology. In general, only impacts data are considered, because the vulnerability and mitigation measures are more complex to incorporate. In this paper the novelty of the approach is to have included vulnerability and mitigation measures in the methodology.

We added the following sentence in the conclusions: "Due to the quality of the data set, we were able to quantify the impact of taking into account or not the information about vulnerability but also different methods for accounting for mitigation measures on the computation of performance criteria." We added this sentence in the conclusion [lines 21-23p.16].

8/ Figure 1 is the same as Figure 4 already published in Lagadec et al., 2018. I understand that you are using the same method. But if a figure has been already published, this should be mentioned in the paper. Moreover, the description of the IRIP method on page 3 uses exactly the same words used in Lagadec et al., 2018. [Lagadec, L.-R., Moulin, L., Braud, I., Chazelle, B., and

- 30 Breil, P.: A surface runoff mapping method for optimizing risk assessment on railways, Safety Science, 110, 253-267, https://doi.org/10.1016/j.ssci.2018.05.014, 2018. ]
  Figure 1 was not exactly the same as the one published in Lagadec et al. (2018). Nevertheless we modified the figure and we
- mention that the figure is "adapted from Lagadec et al. (2018)" in the figure caption. In terms of description of the IRIP method, we have already mentioned in the current version of the paper (p.3 line 19-21) that
- 35 the provided description was mainly borrowed from Lagadec et al. (2018): "The present description is mainly taken from Lagadec et al. (2018) that retained improvements proposed by Lagadec (2017) to the IRIP model." We however modified the description as follows to make it less similar to that of Lagadec et al. (2018), and to include also the answer to Reviewer#2 comment 9/ and Reviewer#1 comment 3.3/ in the description.
- 40 "The IRIP model is briefly described here, but more details can be found in the literature (Dehotin and Breil, 2011; Lagadec et al., 2018). The present description is mainly taken from Lagadec et al. (2018) that retained improvements proposed by Lagadec (2017) to the IRIP model. The IRIP model provides three maps representing three processes involved in storm runoff hazard: generation, transfer and accumulation of runoff. Runoff generation occurs in areas with low infiltration capacity,

[revised manuscript text omitted]

10/ Page 5 lines 15-16: this statement is incorrect. The results of the chi-square do not demonstrate that the relationship is highly significant, but that it possible to reject the null hypothesis of independence, because it is unlikely that the null hypothesis of independence is true.

The sentence was modified as follows:

5 "Thus, a value of  $\chi^2$  larger than 10.83 means that the null hypothesis (independence between the risk levels and the IRIP map) can be rejected at the 0.1% level." [lines 30-31 p.5]

11/Page 10 lines 2-8: the demonstration (or assumption?) that each section had the chance to experience a rare event is obscure to me.

- 10 The point mentioned by Reviewer #2 was raised in the manuscript to support the fact that the chosen case study was adequate to assess the relevance of the proposed evaluation methodology. In particular, the evaluation of the methodology would be biased if the duration of data collection was not long enough so that each section of the railway has had the opportunity to be affected by a heavy rainfall event. In this case, the IRIP model could indicate a risk in a section without reported runoff-related impact because no intense rainfall event would have occurred at that location. To show that the hypothesis that each railway
- 15 section has had an equal opportunity to experience a high rainfall event, we computed the probability of experiencing [resp. not experiencing] rainfall events of several return periods over a duration of 100 years. This probability is  $(1-(1-0.1)^{100}) = 0.99997$  [resp. less than 0.0001%] for a 10-year return period,  $(1-(1-0.05)^{100}) = 0.994$  [resp. less than 1%] for a 20-year return period, and  $(1-(1-0.0.02)^{100}) = 0.867$  [resp. 13%] for a 50-year return period. Therefore, the working hypothesis is valid and we can conclude that our application of the evaluation methodology is not biased and that the case study was adequate

20 to assess the relevance of the proposed evaluation methodology. We reformulated the sentences as follows:

"Furthermore, although it covers more than one century of data, the database may not be comprehensive, which could affect the false alarm ratio if all the occurred impacts have not been recorded. Moreover, the evaluation was conducted assuming (see section 2.3.3) that each section of the railway had the opportunity to be affected by runoff, i.e. that each section of the

- railway had the opportunity to be affected by an intense rainfall event. If it was not the case, the IRIP model could indicate a risk in a section that would not have been impacted in the absence of any intense rainfall event at that location. To assess the validity of this working hypothesis: 'each section had the opportunity to be affected by an intense runoff event', we can calculate the probability of not having experienced a rainfall event of a given return period during one century. This probability is less than 0.001% for a 10-year return period [(1/10)100], less than 1% [(1/20)100] for a 20-year return period, and 13%
- 30 [(1/50)100] for a 50-year return period respectively. Therefore, it can be assumed that each section of the railway had the opportunity to experience a rare event at least once during the data collection period. This shows that, if the database is long enough and of course comprehensive (i.e. all the occurred runoff-related impacts were properly reported), the working hypothesis can be accepted and therefore, performance measures can be considered as not biased. In the present case, the comprehensiveness of the database is exceptional, but far from being perfect. However, it was the best that could be collected,
- 35 and the duration of data collection (more than one century) ensures that the chosen case study was relevant to assess the accuracy of the proposed evaluation methodology." [lines 4-20 p.12]

**Minor comments:**

12/ Page 3 line 39: reduces->reduced Corrected.

40 13/ Page 6 line 15: either...or Corrected.

14/ Table 2: please specify also in the table the number of d.o.f for the chi-square 
[revised manuscript text omitted]

<del>(Hits) + (False alarms)</del> | <del>Varies from 0 to 1</del>
Perfect score:0                                                                                                                                                   |
| <del>χ² test</del>             |                                                          | $\frac{P(\chi^2 \ge 10,83) = 0,001}{\text{``Highly significant ``}}$ $\frac{P(\chi^2 \ge 7.88) = 0,005}{\text{``Very significant ``}}$ $\frac{P(\chi^2 \ge 6.63) = 0,01}{\text{``Significant ``}}$ |

Table 13: Parameterization of the IRIP model for the case study. The table provides values of the thresholds used for each indicator when condition is favorable (score = 1)

l

| IRIP maps    | Indicators          | Thresholds used for favorable conditions (score =1)                               |  |
|--------------|---------------------|-----------------------------------------------------------------------------------|--|
|              | Soil permeability   | Saturated hydraulic conductivity (Ks) $< 10^{-6}$ m s -1 + urban areas |  |
|              | Soil thickness      | Soil thickness < 50 cm + urban areas                                              |  |
|              | Soil slaking        | Urban areas + slacking $\geq 3$                                                   |  |
| Generation   | Soli slaking        | Slacking computed according to Cerdan et al. (2006)                               |  |
|              |                     | Slope > Threshold_1 OR topographic index > Threshold_2                            |  |
|              | Topography          | Threshold_1 and Threshold_2 determined using a classification                     |  |
|              |                     | algorithm (Rubin, 1967)                                                           |  |
|              | Land use            | Urban areas and agricultural lands                                                |  |
|              | Upstream generation | Modal value of the unstream sub-catchment $\geq 3$                                |  |
|              | susceptibility      | $\frac{1}{2}$                                                                     |  |
| -            | Slope               | Slope > Threshold_1                                                               |  |
| Transfer     | Break of slope      | Convex break of slope $\geq 0,0018$                                               |  |
| Transier     | break of slope      | (GRASS GIS r.param.scale function ; 3 pixels)                                     |  |
|              | Drained area        | Drained area $\geq$ 2.5 ha (Lagadec, 2017)                                        |  |
|              | Soil erodibility    | Erodibility - Urban areas + erodibility $\ge 3$                            |  |
|              | Son crodionity      | Erodibility computed according to Cerdan et al. (2006)                            |  |
|              | Upstream generation | Modal value of the unstream sub-catchment $\geq 3$                                |  |
|              | susceptibility      |                                                                                   |  |
|              | Slope               | Slope $\leq$ Threshold_1                                                          |  |
| Accumulation | Break of slope      | Concave break of slope $\leq$ -0,0018                                             |  |
|              | break of slope      | (GRASS GIS r.param.scale function ; 3 pixels)                                     |  |
|              | Topographic index   | Topographic index > Threshold_2                                                   |  |
|              | Drained area        | Drained area > 2.5 ha (Lagadec, 2017)                                             |  |

Table 2: Contingency table

[revised manuscript text omitted]